



**Distributive rainfall/runoff modelling to determine runoff to baseflow**
**proportioning and its impact on the determination of the ecological reserve**
Andrew Watson[1], Jodie Miller[1], Manfred Fink[2], Sven Kralisch[2], Melanie Fleischer[2], and
Willem de Clercq[3]
*1. Department of Earth Sciences, Stellenbosch University, Private Bag X1, Matieland 7602,*
*South Africa*
*2. Department of Geoinformatics, Friedrich-Schiller-University Jena, Loebdergraben 32,*
*07743 Jena, Germany*
*3. Stellenbosch Water Institute, Stellenbosch University, Private Bag X1, Matieland, 7602,*
*South Africa*
**Keywords:** rainfall/runoff modelling, Verlorenvlei, J2000, ecological reserve
**Abstract**
River systems that support high biodiversity profiles are conservation priorities world-wide.
Understanding river eco-system thresholds to low flow conditions is important for the
conservation of these systems. While climatic variations are likely to impact the streamflow
variability of many river courses into the future, understanding specific river flow dynamics
with regard to streamflow variability and aquifer baseflow contributions are central to the
implementation of protection strategies. While streamflow is a measurable quantity, baseflow
has to be estimated or calculated through the incorporation of hydrogeological variables. In
this study, the groundwater components within the J2000 rainfall/runoff model were distributed
to provide daily baseflow and streamflow estimates needed for ecological reserve
determination. The modelling approach was applied to the RAMSAR-listed Verlorenvlei





estuarine lake system on the west coast of South Africa which is under threat due to agricultural
expansion and climatic fluctuations. The sub-catchment consists of four main tributaries, the
Krom Antonies, Hol, Bergvallei and Kruismans. Of these, the Krom Antonies tributary was
initially presumed the largest baseflow contributor, but was shown to have significant
streamflow variability, attributed to the highly conductive nature of the Table Mountain Group
sandstones and quaternary sediments. The Bergvallei tributary was instead identified as the
major contributor of baseflow. The Hol tributary was the least susceptible to streamflow
fluctuations due to the higher baseflow proportion (56%), as well as the dominance of less
conductive Malmesbury shales which underlie this tributary. The estimated flow exceedance
probabilities indicated that during the wet cycle (2007-2017) the average inflow supported the
evaporative demands if the lake was at 40 % capacity, while during the dry cycle (1997-2008),
only 15 % of the lake's capacity would be met. The exceedance probabilities estimated in this
study suggest that inflows from the four main tributaries are not enough to support the lake
during dry cycles, with the evaporation demand of the entire lake being met only 38 % of the
time. This study highlighted the importance of low occurrence events for filling up the lake,
allowing for regeneration of lake supported ecosystems. While the increased length of dry
cycles are likely to result in the lake drying up more frequently, it is important to ensure that
water resources are not overallocated during wet cycles, hindering ecosystem regeneration and
prolonging the length of these dry cycle conditions.



## 1. Introduction


Functioning river systems offer numerous economic and social benefits to society including
water supply, nutrient cycling and disturbance regulation amongst others (e.g Costanza et al.,
1997; Postel and Carpenter, 1997). As a result, many countries worldwide have endeavoured
to protect river ecosystems after provision has been made for basic human needs (eg Rowlston
and Palmer, 2002). However, the implementation of river protection has been problematic
(Richter, 2010), because many river courses and flow regimes have been severely altered due
to socio-economic development. Previously, river health problems were thought to be only a
result of low-flow conditions and therefore, if minimum flows were kept above a critical level,
the river's ecosystem would be protected (Poff et al., 1997). It is now recognised that a more
natural flow regime, which includes floods as well as low and medium flow conditions is
required for sufficient ecosystem functioning (Postel and Richter, 2012). For these reasons,
before protection strategies can be developed or implemented for a river system, a
comprehensive understanding of the river flow regime dynamics is necessary.
River flow regime dynamics include consideration of not just the surface water in the river but
also other water contributions including runoff, interflow and baseflow which are all essential
for the maintenance of the discharge requirements. Taken together these factors all contribute
to the determination of what is called the ecological reserve, the minimum environmental
conditions needed to maintain the ecological health of a river system (Hughes, 2001). A variety
of different methods have been developed to incorporate various river health factors into
ecological reserve determination (Acreman and Dunbar, 2004). One of the simplest and most
widely applied, is where compensation flows are set below reservoirs and weirs, using flow
duration curves to derive mean flow or flow exceedance probabilities (e.g. Souchon and Keith,
2001). This approach focusses purely on hydrological indices, which are rarely ecologically
valid (e.g. Barker and Kirmond, 1998).





More comprehensive methods such as functional analysis are focused on the whole ecosystem,
including both hydraulic and ecological data (e.g. Building Block Methodology: King and
Louw, 1998). While these methods consider that a variety of low, medium and high flow events
are important for maintaining ecosystem diversity, they require specific data regarding the
hydrology and ecology of a river system, which in many cases does not exist, has not been
recorded continuously or for sufficient duration (Acreman and Dunbar, 2004). To speed up
ecological reserve determination, river flow records have been used to analyse natural
seasonality and variability of flows (e.g. Hughes and Hannart, 2003). However, this approach
requires long-term streamflow and baseflow timeseries. Whilst streamflow is a measurable
quantity subject to a gauging station being in place, baseflow has to be modelled based on
hydrological and hydrogeological variables.
While rainfall/runoff models can be used to calculate hydrological variables using distributive
surface water components (e.g. J2000: Krause, 2001), groundwater variables are lumped within
the modelling framework. In contrast, groundwater models which distribute groundwater
variables (e.g. MODFLOW: Harbaugh, Arlen, 2005), are frequently setup to lump climate
components. In order to accurately model daily baseflow, which is needed for ecological
reserve determination, modelling systems need to be setup such that both groundwater and
climate variables are treated in a distributive manner (e.g Kim et al., 2008). Rainfall/runoff
models which use Hydrological Response Units (HRUs) as an entity of homogenous climate,
rainfall, soil and landuse properties (Flügel, 1995) are able to reproduce hydrographs through
model calibration (Wagener and Wheater, 2006). However, they are rarely able to correctly
proportion runoff and baseflow components (e.g. Robson et al., 1992). To correctly determine
groundwater baseflow using rainfall/runoff models such as the J2000, aquifer components need
to be distributed. This can be achieved using net recharge and hydraulic conductivity collected
through aquifer testing or groundwater model values.



To better understand the nature of river flow regime dynamics, a J2000 rainfall/runoff model
previously setup to simulate surface water processes (Watson et al., 2018) was distributed to
incorporate aquifer hydraulic conductivity within model HRUs using calibrated values from a
groundwater model (Watson, submitted). The model was setup for the RAMSAR listed
Verlorenvlei estuarine system on the west coast of South Africa, which is under threat from
climate change and agricultural expansion. While the estuarine lake's importance is well
documented (Martens et al., 1996; Wishart, 2000), the ecological reserves of the main feeding
tributaries have not yet been set, partially due to a lack of streamflow and baseflow estimates
within the sub-catchment. The modelling framework developed in this study aimed to provide
the hydrological components, (baseflow and runoff proportioning) of the tributaries needed to
set the ecological reserve. The surface water and groundwater components of the model were
calibrated for two different tributaries which were believed to be the main source of runoff and
baseflow for the sub-catchment. The baseflow and runoff rates calculated from the model
indicate not only that the lake system cannot be sustained by baseflow during low flow periods
but also that the initial understanding of which tributaries are key to the sustainability of the
lake system was not correct. The results have important implications for how we understand
water dynamics in water stressed catchments and the sustainability of ecological systems in
these environments.
**2. Study site**
Verlorenvlei is an estuarine lake situated on the west coast of South Africa, approximately 150
km north of the metropolitan city of Cape Town (Fig. 1). The west coast, which is situated in
the Western Cape Province of South Africa, is subject to a Mediterranean climate where the
majority of rainfall is received between May to September. The Verlorenvlei lake, which is
approximately 15 km$^2$ in size draining a watershed of 1832 km$^2$, forms the southern sub-





catchment of the Olifants/Doorn quaternary catchment. The estuarine lake supports both
Karroid and Fynbos biomes, due to the intermittent connection between salt and fresh water.
A sandbar created around a sandstone outcrop (Table Mountain Group) allows for freshwater
to exit the lake to the sea, as well as reducing sea water flow within the lake. The lake is
supplied by four main tributaries which are the Krom Antonies, Bergvallei, Hol and Kruismans.
The main freshwater sources are suggested to be the Krom Antonies (Sigidi, 2018) and the
Bergvallei, which drain the mountainous regions to the south (Piketberg) and north of the sub-
catchment respectively. The Hol and Kruismans tributaries are variably saline (Sigidi, 2018),
due to high evaporation rates in the valley. Average daily temperatures during summer within
the sub-catchment are between 20-30 °C, with estimated potential evaporation rates of 4 to 6
mm.d$^{-1}$ (Muche et al., 2018). In comparison, winter daily average temperatures are between 12-
20 °C, with estimated potential evaporation rates of 1 to 3 mm.d$^{-1}$ (Muche et al., 2018).

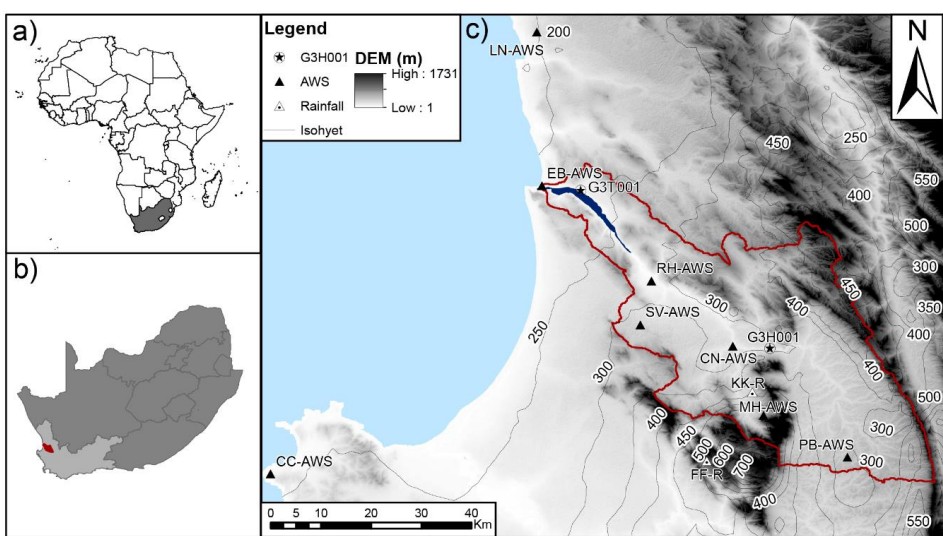


Figure 1: a) Location of South Africa, b) the location of the study catchment within the Western
Cape and c) the extend of the Verlorenvlei sub-catchment with the climate stations, gauging
station (G3H001), measured lake water level (G3T001) and rainfall isohets



The sub-catchment is comprised of three main lithological units, namely: 1) quaternary
sediments 2) Table Mountain Group (TMG) sandstones and 3) Malmesbury (MG) shales (Fig
2). The quaternary sediments make up the primary aquifer within the sub-catchment, while the
TMG sandstones and MG shales make up the secondary aquifer. The catchment valley, which
receives the least mean annual precipitation (MAP) (150-500 mm.year$^{-1}$: Lynch, 2004), is
comprised of quaternary sediments that vary in texture, although the majority of the sediments
in the sub-catchment are sandy in nature. The higher relief mountainous regions of the sub-
catchment, which receive the highest MAP (550-1000 mm.year$^{-1}$: Lynch, 2004), are mainly
comprised of fractured TMG sandstones, (youngest to oldest): Peninsula, Graafwater (not
shown), and Piekernerskloof formations (Fig. 2). Underlying the sandstones and quaternary
sediments are the MG shales, which are comprised of the Mooresberg, Piketberg and
Klipheuwel formations (Fig. 2). The MG shales and quaternary sediments which host the
secondary and primary aquifer respectfully, are frequently used to supplement irrigation during
the summer months of the year. During winter, the majority of the irrigation water needed for
crop growth is supplied by the sub-catchment tributaries or lake itself. Agriculture is the
dominant water user in the sub-catchment with an estimated usage of 20 % of the total recharge
(DWAF, 2003; Watson *et al*, submitted), with the main food crop being potatoes. For further
information regarding the study site refer to Watson *et al*., (2018).


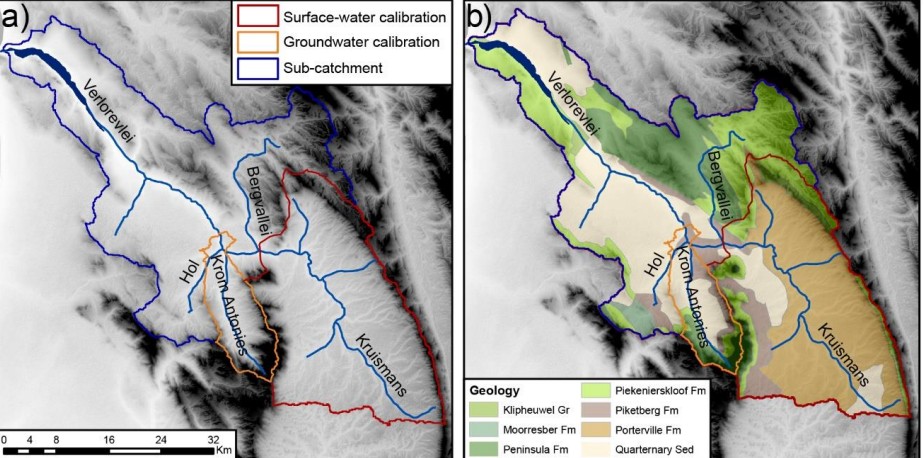


Figure 2: a) The Verlorenvlei sub-catchment with the surface water calibration tributary
(Kruismans) and groundwater calibration tributary (Krom Antonies) and b) the hydrogeology
of the sub-catchment with Malmesbury shale formations (Klipheuwel, Mooresberg, Porterville,
Piketberg), Table Mountain Group formations (Peninsula, Piekenierskloof) and quaternary
sediments
**3. Methodology**
For this study, the J2000 coding was adapted to incorporate distributive groundwater
components. To do this, the hydraulic conductivity or maximum percolation value for specific
geological formations was assigned to the model HRU's, using net recharge and calibrated
aquifer hydraulic conductivity from a groundwater model (Watson, submitted) determined for
eight hydraulic zones (Fig. 3). The adaption was applied to the groundwater components of the
J2000 coding which influenced the portioning of water routed to runoff and baseflow. To
validate the outputs of the model, an empirical mode decomposition (EMD) (Huang et al.,
1998) was applied to compute the proportion of variation in discharge timeseries that attributed
to a high and low water level change at the sub-catchment outlet.





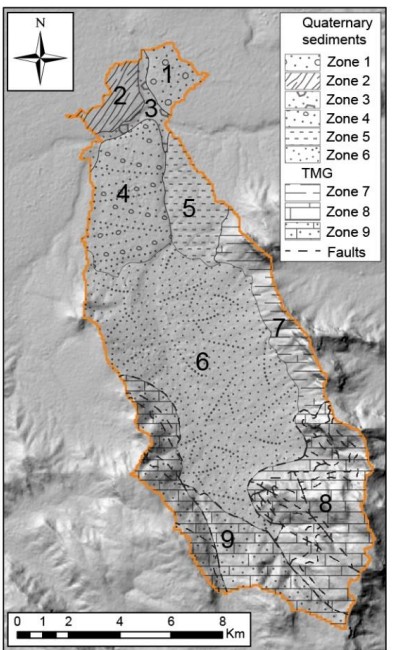


Figure 3: The aquifer hydraulic zones used for the groundwater calibration of the J2000 (after
Watson, submitted)

## 3.1 Hydrological Response Unit Delineation

HRUs and stream segments (reaches) are used within the J2000 model for distributive
topographic and physiological modelling. In this study, the HRU delineation made use of a
digital elevation model, with slope, aspect, solar radiation index, mass balance index and
topographic wetness being derived. Before the delineation process, gaps within the digital
elevation model were filled using a standard fill algorithm from ArcInfo (Jenson and
Domingue, 1988). The AML (ArcMarkupLanguage) automated tool (Pfennig et al., 2009) was
used for the HRU delineation, with between 13 and 14 HRUs/km$^2$ being defined
(Pfannschmidt, 2008). After the delineation of HRUs, dominant soil, land use and geology
properties were assigned to each. The hydrological topology was defined for each HRU by
identifying the adjacent HRUs or stream segments that received water fluxes.



**3.2 Model regionalisation**

Rainfall and relative humidity are the two main parameters that are regionalised within the J2000 rainfall/runoff model. While a direct regionalisation using an inverse-distance method (IDW) and the elevation of each HRU can be applied to rainfall data, the regionalisation of relative humidity requires the calculation of absolute humidity. The regionalisation of rainfall records was applied by defining the number of weather station records available and estimating the influence on the rainfall amount for each HRU. A weighting for each station using the distance of each station to the area of interest was applied to each rainfall record, using an elevation correction factor (Watson et al., 2018). The relative humidity and air temperature measured at set weather stations was used to calculate the absolute humidity. Absolute humidity was thereafter regionalised using the IDW method, station and HRU elevation. After the regionalisation had been applied, the absolute humidity was converted back to relative humidity through calculation of saturated vapor pressure and the maximum humidity.

**3.3 Water balance calculations**

The J2000 model is divided into calculations that impact surface water and groundwater processors. The J2000 model distributes the regionalised precipitation ($P$) calculated for each HRU using a water balance defined as:

$$P = R + Int_{max} + ETR + \Delta Soil_{sat} \tag{1}$$

where $R$ is runoff (mm) (RD1 - surface runoff; RD2 - interflow), $Int_{max}$ is vegetation canopy interception (mm), $ET$R is 'real' evapotranspiration and $\Delta Soil_{sat}$ is change in soil saturation. The surface water processes have an impact on the amount of modelled runoff and interflow, while the groundwater processors influence the upper and lower groundwater flow components.



### 3.3.1 Surface water components

Potential evaporation (ETP) within the J2000 model is calculated using the Penman Monteith equation. Before evaporation was calculated for each HRU, interception was subtracted from precipitation using the leaf area index and leaf storage capacity for vegetation (a_rain) (Supplementary: Table 1). Evaporation within the model considers several variables that influence the overall modelled evaporation. Firstly, evaporation is influenced by a slope factor, which was used to reduce ETP based on a linear function. Secondly, the model assumed that vegetation transpires until a particular soil moisture content where ETP is reached, after which modelled evaporation was reduced proportionally to the ETP, until it became zero at the permanent wilting point.

The soil module in the J2000 model is divided up into processing and storage units. Processing units in the soil module include soil-water infiltration and evapotranspiration, while storage units include middle pore storage (MPS), large pore storage (LPS) and depression storage. The infiltrated precipitation was calculated using the relative saturation of the soil, and its maximum infiltration rate (SoilMaxInfSummer and SoilMaxInfWinter) (Supplementary: Table 1). Surface runoff was generated when the maximum infiltration threshold was exceeded. The amount of water leaving LPS, which can contribute to recharge, was dependant on soil saturation and the filling of LPS via infiltrated precipitation. Net recharge ($R_{net}$) was estimated using the hydraulic conductivity ($SoilMaxPerc$), the outflow from LPS ($LPS_{out}$) and the slope ($slope$) of the HRU according to:

$$R_{net} = LPS_{out} \times (1 - \tan{(slope)} \, SoilMaxPerc) \tag{2}$$

The hydraulic conductivity, $SoilMaxPerc$ and the adjusted $LPS_{out}$ were thereafter used to calculate interflow ($IT_f$) according to:





$$IT_f = LPS_{out} \times (\tan(slope) \, SoilMaxPerc) \qquad (3)$$

with the interflow calculated representing the sub-surface runoff component RD2 and is routed
as runoff within the model.

### 3.3.2 Groundwater components

The J2000 model for the Verlorenvlei sub-catchment was set up with two different geological
reservoirs: (1) the primary aquifer (upper groundwater reservoir - RG1), which consists of
quaternary sediments with a high permeability; and (2) the secondary aquifer (lower
groundwater reservoir- RG2), made up of MG shales and TMG sandstones (Table 1).

| Aquifer | Formation | Type | RG1_max (mm) | RG2_max (mm) | RG1_k (d) | RG2_k (d) | RG1_active (n/a) | Kf_geo (mm/d) | depthRG1 (cm) |
|---|---|---|---|---|---|---|---|---|---|
| Primary | Quarternary Sediments | Sediments | 50 | 700 | 100 | 431 | 1 | 500 | 1750 |
| Secondary/MG | Moorresberg Formation | Shale Greywacke | 0 | 580 | 0 | 350 | 0 | 950 | 1750 |
| Secondary/MG | Porterville Formation | Shale Greywacke | 0 | 560 | 0 | 335 | 0 | 2 | 1750 |
| Secondary/MG | Piketberg Formation | Shale Greywacke | 0 | 1000 | 0 | 600 | 0 | 950 | 1750 |
| Secondary/MG | Klipheuwel Group | Shale Greywacke | 0 | 500 | 0 | 300 | 0 | 950 | 1750 |
| Secondary/TMG | Peninsula Formation | Sandstone | 0 | 1000 | 0 | 600 | 0 | 950 | 1750 |
| Secondary/TMG | Piekenierskloof Formation | Sandstone | 0 | 600 | 0 | 400 | 0 | 1 | 1750 |

Table 1: The J2000 hydrogeological parameters RG1_max, RG2_max, RG1_k, RG2_Kf_geo
and depthRG1 assigned to the primary and secondary aquifer formations for the Verlorenvlei
sub-catchment
The model therefore considered two baseflow components, a fast one from the RG1 and a
slower one from RG2. The filling of the groundwater reservoirs was done by net recharge, with
emptying of the reservoirs possible by lateral subterranean runoff as well as capillary action in
the unsaturated zone. Each groundwater reservoir was parameterised separately using the
maximum storage capacity (maxRG1 and maxRG2) and the retention coefficients for each
reservoir ($recRG1$ and $recRG2$). The outflow from the reservoirs was determined as a function
of the actual filling ($actRG1$ and $actRG2$) of the reservoirs and a linear drain function.
Calibration parameters $recRG1$ and $recRG2$ are storage residence time parameters. The
outflow from each reservoir was defined as:



$$OutRG1 = \frac{1}{gwRG1Fact \times recRG1} \times actRG1 \qquad (4)$$

$$OutRG2 = \frac{1}{gwRG2Fact \times recRG2} \times actRG2 \qquad (5)$$

where $OutRG1$ is the outflow from the upper reservoir, $OutRG2$ is the outflow from the lower
reservoir and $gwRG1Fact$/ $gwRG2Fact$ are calibration parameters for the upper and lower
reservoir used to determine the outflow from each reservoir. To allocate the quantity of net
recharge between the upper (RG1) and lower (RG2) groundwater reservoirs, a calibration
coefficient $gwRG1RG2sdist$ was used to distribute the net recharge for each HRU using the
HRU slope. The influx of groundwater into the shallow reservoir ($inRG1$) was defined as:

$$inRG1 = R_{net} \times (1 - (1 - \tan(slope))) \times gwRG1RG2sdist \qquad (6)$$

The influx of net recharge into the lower groundwater reservoir ($inRG2$) was defined as:

$$inRG2 = R_{net} \times (1 - \tan(slope)) \times gwRG1RG2sdist \qquad (7)$$

with the combination of $OutRG1$ and $OutRG2$ representing the baseflow component that is
routed as an outflow from the model.
**3.4 Lateral and reach routing**
Lateral routing was responsible for water transfer within the model and included HRU influxes
and discharge through routing of cascading HRUs from the upper catchment to the exit stream.
HRUs were either able to drain into multiple receiving HRUs or into reach segments, where
the topographic ID within the HRU dataset determined the drain order. The reach routing
module was used to determine the flow within the channels of the river using the kinematic
wave equation and calculations of flow according to Manning and Strickler. The river
discharge was determined using the roughness coefficient of the stream (Manning roughness),
the slope and width of the river channel and calculations of flow velocity and hydraulic radius
calculated during model simulations.



### 4.1 J2000 Input data

After the above adaptations of the J2000 model coding, input data representing both the surface

water and groundwater components were required.

#### 4.1.1 Surface water components

Climate and rainfall: Rainfall, windspeed, relative humidity, solar radiation and air temperature

were monitored by Automated Weather Stations (AWS) within and outside of the study

catchment (Fig. 1). Of the climate and rainfall data used during the surface water modelling

(Watson et al., 2018), data was sourced from six AWS's of which four stations were owned by

the South African Weather Service (SAWS) and three by the Agricultural Research Council

(ARC). Two stations that were installed for the surface water modelling, namely Moutonshoek

(M-AWS) and Confluence (CN-AWS) were used for climate and rainfall validation due to their

short record length. Additional rainfall data collected by farmers at high elevation at location

FF-R and within the middle of the catchment at KK-R were used to improve the climate and

rainfall network density.

Landuse classification: The vegetation and landuse dataset that was used for the sub-catchment

(CSIR, 2009) included five different landuse classes: 1) wetlands and waterbodies, 2)

cultivated (temporary, commercial, dryland), 3) shrubland and low fynbos, 4) thicket,

bushveld, bush clumps and high fynbos and 5) cultivated (permanent, commercial, irrigated).

Each different landuse class was assigned an albedo, root depth and seal grade value based on

previous studies (Steudel et al., 2015)(Supplementary: Table 2). The Leaf Area Index (LAI)

and vegetation height varies by growing season with different values of each for the particular

growing season. While surface resistance of the landuse varied monthly within the model, the

values only vary significantly between growing seasons.





Soil dataset: The Harmonized World Soil Database (HWSD) v1.2 (Batjes et al., 2012) was the
input soil dataset, with nine different soil forms within the sub-catchment (Supplementary:
Table 3). Within the HWSD, soil depth, soil texture and granulometry were used to calculate
and assign soil parameters within the J2000 model. MPS and LPS which differ in terms of the
soil structure and pore size were determined in Watson et al. (2018), using pedotransfer
functions within the HYDRUS model (Supplementary: Table 3).
Streamflow and water levels: Streamflow, measured at the Department of Water Affairs
(DWA) gauging station G3H001 between 1970-2009, at the outlet of the Kruismans tributary
(Het Kruis) (Fig 1 and 2), was used for surface water calibration. The G3H001 two-stage weir
could record a maximum flow rate of 3.675 $m^2.s^{-1}$ due to the capacity limitations of the
structure. After 2009, the G3H001 structure was decommissioned due to structural damage,
although repairs are expected in the near future due to increasing concerns regarding the influx
of freshwater into the lake. Water levels measured at the sub-catchment outlet at DWA station
G3T001 (Fig 1) between 1994 to 2018 were used for EMD filtering.
*4.1.2 Groundwater components*
Net recharge and hydraulic conductivity: The net recharge and hydraulic conductivity values
used for the groundwater model calibration were collected from detailed MODFLOW
modelling conducted for the Krom Antonies tributary (Fig. 3) (Watson, submitted). The net
recharge and aquifer hydraulic conductivity for the Krom Antonies tributary, was estimated
through PEST autocalibration using hydraulic conductivities from previous studies (SRK,
2009; UMVOTO-SRK, 2000) and potential recharge estimates (Watson et al., 2018).
Hydrogeology: Within the hydrogeological dataset, parameters assigned include maximum
storage capacity (RG1 and RG2), storage coefficients (RG1 and RG2), the minimum
permeability/maximum percolation (Kf_geo of RG1 and RG2) and depth of the upper





groundwater reservoir (depthRG1). The maximum storage capacity was determined using an
average thickness of each aquifer and the total number of voids and cavities, where the primary
aquifer thickness was assumed to be between 15-20 m (Conrad et al., 2004), and the secondary
aquifer between 80-200 m (SRK, 2009). The maximum percolation of the different geological
formations was assigned hydraulic conductivities using the groundwater model for the Krom
Antonies sub-catchment (Watson et al., submitted). The J2000 geological formations were
assigned conductivities to modify the maximum percolation value to ensure internal
consistency with recharge values calculated using MODFLOW (Table 1).
**4.2 Model calibration**
*4.2.1 Model sensitivity*
The J2000 sensitivity analysis for Verlorenvlei sub-catchment was presented in Watson *et al*.,
(2018) and therefore only a short summary is presented here. In this study, parameters that
were used to control the ratio of interflow to percolation were adjusted, which in the J2000
model include a slope (SoilLatVertDist) and max percolation value. The sensitivity analysis
conducted by Watson *et al*., (2018) showed that for high flow conditions (E2) (Nash-Sutcliffe
efficiency in its standard squared), model outputs are most sensitive to the slope factor, while
for low flow conditions (E1) (modified Nash-Sutcliffe efficiency in a linear form) the model
outputs were most sensitive to the maximum infiltration rate of the soil (ie. the parameter
maxInfiltrationWet) (Supplementary: Figure 1). The max percolation was moderately sensitive
during wet and dry conditions, and together with the slope factor, controlled the interflow to
percolation portioning that was calibrated in this study.
*4.2.2 Surface water calibration*
The surface water parameters of the model were calibrated for the Kruismans tributary (688
km$^2$) (Fig. 2) using the gauging data from G3H001 (Fig. 4 and Table 1). The streamflow data



336 used for the calibration was between 1986-1993, with model validation between 1994 to 2007

337 (Fig. 4). This specific calibration period was selected due to the wide range of different runoff

338 conditions experienced at the station, with both low and high flow events being recorded. For

339 the calibration, the modelled discharge was manipulated in the same fashion, with a maximum

340 value of 3.675 m³/s, so that the tributary streamflow behaved as measured discharge.

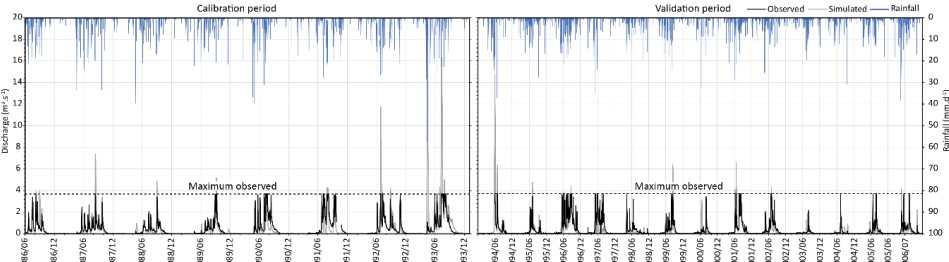

341

342 Figure 4: The surface water calibration (1986-1993) and validation (1986-2006) of the J2000

343 model using gauging data from the G3H001

344 An automated model calibration was performed using the "Nondominating Sorting Genetic

345 Algorithm II" (NSGA-II) multi-objective optimisation method (Deb et al., 2002) with 1023

346 model runs being performed. Narrow ranges of calibration parameters (FC_Adaptation,

347 AC_Adaptation, soilMAXDPS, gwRG1Fact and gwRG2Fact) were chosen to (1) ensure that

348 the modelled recharge from J2000 was within an order of magnitude of recharge from the

349 MODFLOW model; (2) to achieve a representative sub-catchment hydrograph. As objective

350 functions, the E2, E1and the average bias in % (Pbias) were utilized for the calibration (Krause

351 et al., 2005) (Table 2). The choice of the optimized parameter set was made to ensure that E2

352 was better than 0.57 (best value was 0.574302) and the Pbias better than 5% (Table 1). From

353 the automated calibration, 308 parameter sets were determined with the best E1 being chosen

354 to ensure that the model is representative of low flow conditions (Table 1).





### 4.2.3 Model validation


For the surface water model validation, the streamflow records between 1994-2007 were used,
where absolute values (E1) and squared differences (E2) of the Nash Sutcliffe efficiency were
reported. The Pbias was also used as an objective function to report the model performance by
comparison between measured and modelled streamflow (Table 2). Although gauging station
limitations resulted in good objective functions from the model, the performance of objective
functions E1, E2, Pbais reduced between the validation and calibration period (Table 2). During
the calibration period there was a good fit between modelled and measured streamflow (Pbias=-
1.82), with a significant difference between modelled and measured streamflow during the
validation period (Pbias=-19.2). The calibration was performed over a wet cycle (1986-1997),
which resulted in a more common occurrence of streamflow events that exceeded 3.675 $m^3.s^{-1}$
$^1$, thereby reducing the number of calibration points. In contrast the validation was performed
over a dry cycle (1997-2007), which resulted in more data points as few streamflow events
exceeded 3.675 $m^3.s^{-1}$.

|       | Calibration: 1987-1993 | Validation: 1994-2007 |
|-------|------------------------|-----------------------|
| e1    | 0.55085                | 0.53312               |
| e2    | 0.57156                | 0.55736               |
| $R^2$ | 0.61788                | 0.58067               |
| Pbias | -1.82301               | -19.23758             |


Table 2: The objective functions E1, E2, coefficient of determination $R^2$ and Pbias used for the
surface water calibration (198701993) and validation (1987-2007)
The groundwater recharge values from MODFLOW were validated with J2000 recharge
estimates (Fig. 5). During the calibration period the groundwater recharge proportion for the
eight calibrated hydraulic zones (Fig. 3) achieved a good fit, with an average value from J2000
of 4.71 % and from MODFLOW of 4.58 %. The coefficient of determination ($R^2$) between the



J2000 and MODFLOW was 0.81. Across the entire dataset J2000 overestimates groundwater
recharge by 2.75 %, although the coefficient of determination produced an $R^2$ of 0.92 which is
higher than the calibration period.

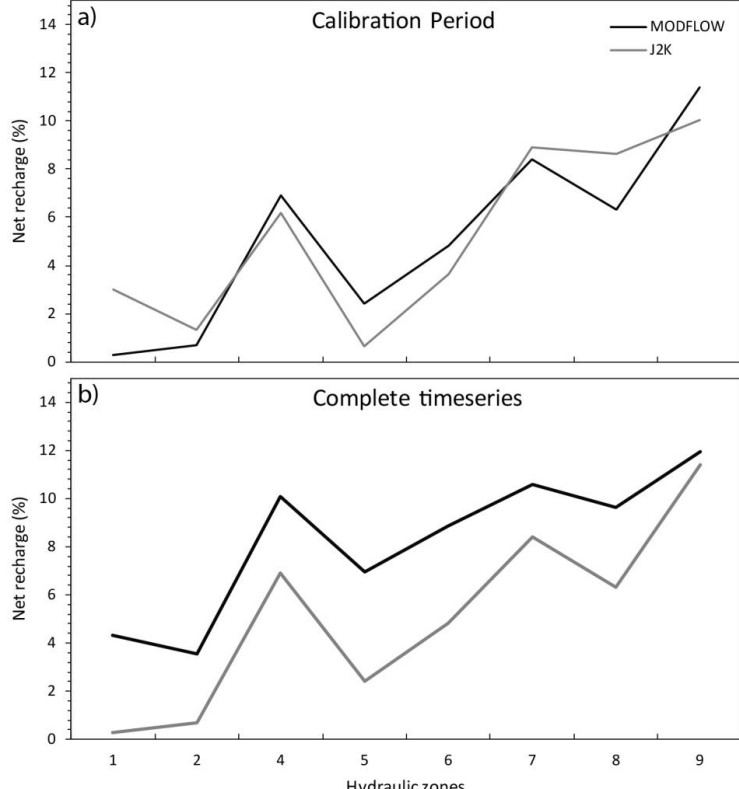


Figure 5: The groundwater calibration for each hydraulic zone with a) net recharge for the
J2000 and MODFLOW during the model calibration (2016) and b) the net recharge deviation
between MODFLOW and J2000 across the entire modelling timestep (1986-2017)
**4.3 EMD filtering**
To account for missing streamflow data between 2007-2017, an Empirical Mode
Decomposition (EMD) (Huang et al., 1998) was applied to the measured water level data at
the sub-catchment outlet (G3T001)(Fig. 1) between 1994 to 2018 (Fig 6a). EMD is a method



for the decomposition of nonlinear and nonstationary signals into sub-signals of varying
frequency, so-called intrinsic mode functions (IMF), and a residuum signal. By removing one
or more IMF or the residuum signal, certain frequencies (e.g. noise) or an underlying trend can
be removed from the original time series data. This approach was successfully applied to the
analysis of river runoff data (Huang et al., 2009) and forecasting of hydrological time series
(Kisi et al., 2014). In this study, EMD filtering was used to remove high frequency sub-signals
from simulated runoff and measured water level data to compare the more general seasonal
variations of both signals (Fig. 6b).






Figure 6: a) The water level fluctuations at station G3T001 with modelled runoff and b) the
EMD filtering showing the variation in discharge timeseries attributed a water level change at
the station





## 5. Results


The J2000 model was used to simulate both runoff and baseflow, with runoff being comprised
of direct surface runoff (RD1) and interflow (RD2) and baseflow simulated from the primary
(RG1) and secondary aquifer (RG2). Below, the results of the modelled streamflow and
baseflow are presented, along with the total flow contribution of each tributary, the runoff to
baseflow proportioning and stream exceedance probabilities. The coefficient of variation (CV)
was used to determine the streamflow variability of each tributary, while the baseflow index
(BFI) was used to determine the baseflow and runoff proportion.

**5.1 Streamflow and baseflow**


Streamflow for the sub-catchment shows two distinctively wet periods (1987-1997 and 2007-
2017), separated by a dry period (1997-2007) (Fig. 7). Yearly sub-catchment rainfall volumes
between 1987-1997 were between 288 and 492 mm/yr$^{-1}$, with an average of 404 mm.yr$^{-1}$. For
this period, average yearly streamflow between 1987-1997 was 1.4 m$^3$.s$^{-1}$, with an average
baseflow contribution of 0.63 m$^3$.s$^{-1}$. The modelled streamflow reached a maximum of 48 m$^3$.s$^{-1}$
in 1993, when 5 m$^3$.s$^{-1}$ of baseflow was generated after 58 mm of rainfall was received.
Between 1997-2007 (dry period) sub-catchment yearly rainfall was between 222 and 394
mm/yr$^{-1}$ with an average of 330 mm.yr$^{-1}$ (Fig. 7). For this period, average yearly streamflow
between 1997-2007 was 0.44 m$^3$.s$^{-1}$, with an average baseflow contribution of 0.18 m$^3$.s$^{-1}$. The
modelled streamflow reached a maximum of 11 m$^3$.s$^{-1}$ in 2002, with a baseflow contribution
of 2.5 m$^3$.s$^{-1}$ after 28 mm of rainfall was received. Between 2007-2017 (wet period) sub-
catchment yearly rainfall was between 231 and 582 mm.yr$^{-1}$ with an average of 427 mm.yr$^{-1}$
(Fig. 7). Over this period, average yearly streamflow between 2007-2017 was 2.5 m$^3$.s$^{-1}$ with
an average baseflow contribution of 1.3 m$^3$.s$^{-1}$. The modelled streamflow reached a maximum



of 52 m$^3$.s$^{-1}$ in 2008, with 13 m$^3$.s$^{-1}$ of baseflow generated after two consecutive rainfall events
each of 25 mm.







Figure 7: a) The average sub-catchment rainfall between 1987-2017 showing wet cycles (1987-1997 and 2008-2017), the modelled streamflow and baseflow inflows for the b) Verlorenvlei, c) Bergvallei, d) Kruismans, e) Krom Antonies and f) Hol with estimated BFI, CV, RD1/RD2, RG1/RG2

## 5.2 Tributary contributions

The four main feeding tributaries (Bergvallei, Kruismans, Hol and Krom Antonies) together contribute 81% of streamflow for the Verlorenvlei, with the additional 19% from small tributaries near Redelinghuys (Fig. 7). The Kruismans contributes most of the total streamflow with 32.4 %, although due to the sub-catchment being the largest of the tributaries (688 km$^2$), the area weighted contribution is 16.4 % (Fig. 7). The Bergvallei (320 km$^2$), which is smaller than the Kruismans, contributes 29 % of the total flow with an area weighted contribution of 32 %. The Krom Antonies has the largest area weighted contribution of 33 % due to its small size (140 km$^2$) in comparison to the other tributaries, although the Krom Antonies contributes only 13 % of the total flow (Fig. 7). The Hol (126 km$^2$) contributes the least total flow with 6.79 %, with a weighted contribution of 18.69 % (Fig. 7).

## 5.3 Flow variability

Streamflow that enters Verlorenvlei has a large daily variability with a coefficient of variation of 189.90 (Fig. 7). Verlorenvlei's streamflow is mainly comprised of surface runoff (RD1/RD2=15.6) as opposed to interflow. The total groundwater flow contribution for the Verlorenvlei is 47 % (BFI=0.47) with the majority of groundwater baseflow from the secondary aquifer (RG1/RG2=0.29). The Kruismans has large daily streamflow variability with a CV of 217.20 (Fig. 7). The Kruismans tributary is mainly comprised of surface runoff (RD1/RD2=13.9) with a small interflow contribution. The total groundwater flow contribution for the Kruismans tributary is relatively low, with groundwater making up 14 % of streamflow





(BFI=0.14), where the majority of baseflow is from the secondary aquifer (RG1/RG2=0.37).
The Bergvallei has the highest streamflow variability, with a CV of 284.54, and the highest
surface runoff to interflow proportion (RD1/RD2=22.50), with a total groundwater
contribution of 49 % (BFI=0.49) (Fig. 7). The secondary aquifer contributes the majority of
baseflow for the Bergvallei, with the secondary aquifer contribution being more than double
the primary aquifer (RG1/RG2=0.54). The Krom Antonies has significant variability in daily
streamflow, with a CV of 283.00 (Fig. 7). The runoff from the Krom Antonies is mainly
comprised of surface runoff with interflow being a minor contribution (RD1/RD2=14.30). The
Krom Antonies has a relatively high groundwater component (BFI=0.34), with the secondary
aquifer contributing the most baseflow (RG1/RG2=0.33). The Hol tributary has the lowest
variability in daily streamflow with a CV of 146.54 (Fig. 7). The Hol tributary is mainly
comprised of surface runoff (RD1/RD2=9.40), although the interflow is the highest proportion
within the sub-catchment. The Hol tributary is mainly comprised of groundwater (BFI=0.56),
with the majority of baseflow being derived from the secondary aquifer (RG1/RG2=0.17).
**5.4 Flow exceedance probabilities**
The results for the flow exceedance probabilities includes flow volumes for each tributary and
the lake's inflow which are exceeded 95%, 75%, 50%, 25 and 5 % of the time. The 95 percentile
corresponds to a lake inflow of 0.054 $m^3.s^{-1}$ or 4,702 $m^3.d^{-1}$, with between 0.001-0.004 $m^3.s^{-1}$
from the feeding tributaries (Table 3). The 75-percentile flow, which is exceeded 3/4 of the
time corresponds to an inflow of 0.119 $m^3.s^{-1}$ or 10,303 $m^3.d^{-1}$, with between 0.005-0.015 $m^3.s^{-}$
$^1$ from the feeding tributaries. Average (50 percentile) streamflow flowing into the Verlorenvlei
is 0.237 $m^3.s^{-1}$ or 20,498 $m^3.d^{-1}$, with between 0.012-0.012 $m^3.s^{-1}$ from the feeding tributaries.
The 25-percentile flow, which is exceeded ¼ of the time corresponds to a lake inflow of 1,067
$m^3.s^{-1}$ or 92,204 $m^3.d^{-1}$ with between 0.044-0.291 $m^3.s^{-1}$ from the feeding tributaries. The lake



inflows that are exceeded 5 % of the time correspond to 6.939 $m^3.s^{-1}$ or 599,535 $m^3.d^{-1}$ with
between 0.224-2.49 $m^3.s^{-1}$ from the feeding tributaries.

| Exceedance percentile | Rainfall mm/yr⁻¹ | Verlorenvlei m³.s⁻¹ | Verlorenvlei m³.d⁻¹ | Kruismans m³.s⁻¹ | Kruismans m³.d⁻¹ | Bergvallei m³.s⁻¹ | Bergvallei m³.d⁻¹ | Krom Antonies m³.s⁻¹ | Krom Antonies m³.d⁻¹ | Hol m³.s⁻¹ | Hol m³.d⁻¹ |
|---|---|---|---|---|---|---|---|---|---|---|---|
| 95 | 227 | 0.054 | 4702 | 0.004 | 346 | 0.001 | 69 | 0.001 | 109 | 0.002 | 176 |
| 90 | 264 | 0.074 | 6356 | 0.007 | 604 | 0.002 | 191 | 0.003 | 232 | 0.003 | 269 |
| 85 | 282 | 0.088 | 7628 | 0.010 | 830 | 0.004 | 366 | 0.004 | 319 | 0.004 | 353 |
| 80 | 290 | 0.104 | 8979 | 0.012 | 1072 | 0.007 | 596 | 0.005 | 392 | 0.005 | 434 |
| 75 | 296 | 0.119 | 10303 | 0.015 | 1291 | 0.010 | 839 | 0.005 | 459 | 0.006 | 508 |
| 70 | 324 | 0.136 | 11759 | 0.018 | 1517 | 0.013 | 1104 | 0.006 | 534 | 0.007 | 587 |
| 65 | 357 | 0.155 | 13373 | 0.021 | 1791 | 0.016 | 1381 | 0.007 | 602 | 0.008 | 676 |
| 60 | 387 | 0.176 | 15180 | 0.024 | 2104 | 0.019 | 1657 | 0.008 | 685 | 0.009 | 786 |
| 55 | 396 | 0.203 | 17575 | 0.029 | 2506 | 0.023 | 1965 | 0.009 | 772 | 0.011 | 913 |
| 50 | 405 | 0.237 | 20498 | 0.035 | 3032 | 0.027 | 2309 | 0.010 | 882 | 0.012 | 1058 |
| 45 | 422 | 0.286 | 24669 | 0.043 | 3755 | 0.032 | 2807 | 0.012 | 1024 | 0.014 | 1222 |
| 40 | 430 | 0.371 | 32023 | 0.058 | 5022 | 0.041 | 3511 | 0.015 | 1258 | 0.017 | 1439 |
| 35 | 437 | 0.516 | 44598 | 0.089 | 7699 | 0.053 | 4613 | 0.020 | 1745 | 0.021 | 1790 |
| 30 | 444 | 0.710 | 61310 | 0.156 | 13511 | 0.076 | 6599 | 0.033 | 2824 | 0.029 | 2481 |
| 25 | 454 | 1.067 | 92204 | 0.291 | 25182 | 0.123 | 10619 | 0.062 | 5387 | 0.044 | 3814 |
| 20 | 481 | 1.571 | 135726 | 0.489 | 42242 | 0.223 | 19295 | 0.110 | 9511 | 0.065 | 5655 |
| 15 | 498 | 2.399 | 207275 | 0.780 | 67408 | 0.421 | 36354 | 0.192 | 16594 | 0.096 | 8262 |
| 10 | 537 | 3.759 | 324746 | 1.324 | 114432 | 0.885 | 76477 | 0.359 | 31045 | 0.141 | 12191 |
| 5 | 575 | 6.939 | 599535 | 2.490 | 215152 | 1.884 | 162795 | 0.929 | 80305 | 0.224 | 19312 |


Table 3: The exceedance probabilities for sub-catchment rainfall and Verlorenvlei, Kruismans,
Bergvallei, Krom Antonies and Hol streamflow in $m^3.s^{-1}$ and $m^3.d^{-1}$

## 6.   Discussion

### 6.1 Modelling in sub-Saharan Africa

A major limitation facing the development and construction of comprehensive modelling
systems in sub-Saharan Africa is the availability of appropriate climate and streamflow data.
For this study, while there was access to over 20 years of streamflow records, the station was
only able to measure a maximum of 3.675 $m^3.s^{-1}$, which hindered calibration of the model for
high flow events. As such, the confidence in the model's ability to simulate high streamflow
events using climate records is limited. While the availability of measured data is a limitation





that could affect the modelled streamflow, discontinuous climate records also hindered the
estimations of long time series streamflow.
Over the course of the 30-year modelling period, a number of climate stations used for
regionalisation were decommissioned and were replaced by stations in different areas. This
required climate regionalisation adaption for simulations over the entire 30-year period to
incorporate the measured streamflow from the gauging station. To account for missing
streamflow records since 2007, an EMD filtering protocol was applied to the runoff data (Fig.
6). The results from the EMD filtering showed that after removing the first nine IMFs, the local
maxima of both signals match the seasonal water level maxima during most of the years. While
considerable improvement can be made to the EMD filtering, the results show some agreement
which suggested that the simulated runoff was representative of inflows into the lake.
In data scares catchments it is important to make use of all available data in an effort to improve
the understanding of the catchment dynamics. To account for historical gauging data a number
of adaptions were made to the climate regionalisation, as well as an EMD filtering protocol to
use water level data at the sub-catchment outlet. Consequently, the model performed
particularly well considering the streamflow and climate station limitations, although the model
is yet to be tested regarding its ability to simulate high flow events.
**6.2 Catchment dynamics**
Factors that impact on streamflow variability are important for understanding river flow regime
dynamics. Previously, factors that affect streamflow variability such as CV and BFI values
have been used to determine how susceptible particular river systems are to drought (e.g
Hughes and Hannart, 2003). While CV values have been used to account for climatic impacts
such as dry and wet cycles, BFI values are associated with runoff generation processes that
impact the catchment. For South African river systems, BFI values are generally below 1





510 implying that runoff exceeds baseflow. In comparison CV values can be in excess of 10

511 implying high variability in streamflow volumes (Hughes and Hannart, 2003). Generally, CV

512 and BFI measures have been applied to quaternary river systems in southern Africa. For this

513 study, these two measurements have been used to understand river flow dynamics in much

514 smaller tributaries.

515 The highest proportion of streamflow needed to sustain the Verlorenvlei lake water level is

516 received from the Bergvallei tributary, although the area weighted contribution from the Krom

517 Antonies is more significant (Fig. 7). However, CV values for the Bergvallei indicate high

518 streamflow variability. This is partially due to the high surface runoff component in modelled

519 streamflow within the Bergvallei in comparison to the minor interflow contribution, suggesting

520 little sub-surface runoff. While streamflow from the Bergvallei tributary is 47% groundwater,

521 which would suggest a more sustained streamflow, due to the TMG dominance as well as a

522 high primary aquifer contribution, baseflow from the Bergvallei is driven by highly conductive

523 rock and sediment materials. Similarly, CV values for the Krom Antonies indicate high

524 streamflow variability due to the presence of a high baseflow contribution from the conductive

525 TMG and primary aquifers. Although the Krom Antonies has a larger interflow component,

526 which would reduce streamflow variability, the dominant TMG presence within this tributary

527 partially compensates for the subsurface flow contributions.

528 In contrast, the Hol has a much smaller daily streamflow variability in comparison to both the

529 Bergvallei and the Krom Antonies (Fig. 7). While streamflow from the Hol tributary is mainly

530 comprised of baseflow (56%), the dominance of low conductive shale rock formations as well

531 as a large interflow component result in reduced streamflow variability. While the larger shale

532 dominance in this tributary not only results in a more sustained baseflow from the secondary

533 aquifer, it also results in large interflow due to the limited conductivity of the shale formations.

534 Compounding the more sustained baseflow from the Hol tributary, the reduced presence of the



primary aquifer results in a dominance in slow groundwater flow from this tributary. Similarly,
the Kruismans is dominated by shale formations which result in a larger interflow contribution,
although due to the limited baseflow contribution (14%) the streamflow from this tributary is
highly variable, which impacts on its susceptibility to drought.
The results from this study have shown that while the Krom Antonies was initially believed to
be the major flow contributor, the Bergvallei is in fact the most significant, although
streamflow from the four tributaries is highly variable, with baseflow from the Hol tributary
the only constant input source. The presence of conductive TMG sandstones and quaternary
sediments in both the Krom Antonies and Bergvallei result in quick baseflow responses with
little flow attenuation. The potential implication of a constant source of groundwater being
provided from the Hol tributary, is that if the groundwater is of poor quality this would result
in a constant input of saline groundwater, with the Krom Antonies and Bergvallei providing
freshwater after sufficient rainfall.
**6.3 Baseflow comparison**
The groundwater components of the J2000 model were adjusted using calibrated net recharge
and aquifer hydraulic conductivity from a MODFLOW model of one of the main feeding
tributaries of the Verlorenvlei. The Krom Antonies was selected for this calibration as it was
previously believed to be the largest input of groundwater to Verlorenvlei (Fig. 2). Baseflow
for the Krom Antonies tributary was previously calculated using a MODFLOW model (Watson
*et al*., 2018), by considering aquifer hydraulic conductivity and average groundwater recharge.
Due to the fact that average recharge was used, baseflow estimates from MODFLOW are likely
to fall on the upper end of daily baseflow values. For the Krom Antonies sub-catchment Watson
et al.*, (2018) estimated baseflow between 14,000 to 19,000 $m^3.d^{-1}$ for 2010-2016. In this study,




similar daily estimates were only exceeded 10 % of the time, with average estimates (50%) of
1,036 $m^3.d^{-1}$ over course of the modelling period (Fig. 7).
Watson et al., (2018) estimates were applied over the course of a wet cycle (2016), and in
comparison average baseflow from J2000 for 2016 was 8, 214 $m^3.d^{-1}$. The daily timestep nature
of the J2000 is likely to result in far lower baseflow estimates, as recharge is only received over
a 6-month period as opposed to a yearly average estimate. The possible implication of this is
that while common groundwater abstraction scenarios have been based on yearly recharge,
abstraction is likely to exceed sustainable volumes during dry months or dry cycles which could
hinder the ability of the aquifer to supply baseflow. While the groundwater components of the
J2000 have been distributed to allow for improved baseflow estimates, the groundwater
calibration was applied to the Krom Antonies. However, this study showed that Bergvallei has
been identified as the largest water contributor. In hind sight, the use of geochemistry to
identify dominant particular tributaries could have aided the groundwater calibration. While it
would have been beneficial to calibrate the groundwater components of the J2000 using the
Bergvallei, incorporating one tributary that is dominated by TMG outcrops and one by shale
would have improved the representativeness of the baseflow estimates from the model. While
the distribution of aquifer components improved modelled baseflow, including groundwater
abstraction scenarios in baseflow modelling in the sub-catchment is important for future water
management for this ecologically significant area.
**6.4 Ecological reserve and evaporative demand**
Exceedance probabilities have been used as approximate estimates of minimum river flow
requirements. The exceedance percentiles used for ecological reserve determination are
streamflow values that are exceeded 95 % of the time (Barker and Kirmond, 1998). For this
study, exceedance probabilities were estimated through rainfall/runoff modelling for the

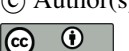



previous 30 years within the Verlorenvlei sub-catchment. The exceedance probabilities were
determined for each tributary, as well as the total inflows into the lake. These exceedance
probabilities were compared with the evaporative demand of the lake, to understand whether
inflows are in surplus or whether evaporation demands exceed inflows. As an approximation
of the evaporative demand of the Verlorenvlei, an average evaporation loss of 5 mm.d$^{-1}$ was
assumed across the lake's surface area (15 km$^2$).
The 95th percentile streamflow contribution, which is the ecological reserve percentile,
corresponds to a lake inflow of 4,702 m$^3$.d$^{-1}$, meeting the evaporation demand if the lake was
at 7 % capacity. From this it does not seem that the 95th percentile is enough to balance the
evaporation demand of the lake. Furthermore, an average streamflow (50th percentile) would
only meet the evaporation demand of 1/4 of the lake's surface area. Considering the exceedance
probability of the wet cycle period (2007-2017), the 95th percentile corresponded to 7,093
m$^3$.d$^{-1}$, meeting the evaporation demand if the lake was at 10% capacity, while an average
inflow (50 percentile) would meet demands if the lake was at 40 % capacity. In contrast, for
the dry cycle (1997-2007), the 95 percentile would correspond to a streamflow of 3,438 m$^3$.d$^{-}$
$^1$, meeting the demand if the lake was at 5 % capacity, while on average (50 percentile) the
demands of 15 % of the lake's capacity would be met.
From the exceedance probabilities generated in this study, the lake is predominately fed by less
frequent large discharge events, where on average the daily inflows to the lake do not sustain
the water level above 40 % capacity. This is particularly evident in the measured water level
data from station G3T001, where measured water levels have a large daily standard deviation
(0.62) (Watson *et al*., 2018). With climate change likely to impact the length and severity of
dry cycles, it is likely that the lake will dry up more frequently into the future, which could
have severe implications on the biodiversity that relies on the lake's habitat for survival. Of
importance to the lake's survival is the protection of river inflows during wet cycles, where the



lake requires these inflows for regeneration and the overallocation of resources could result in
prolonged dry cycle conditions.

## 7. Conclusion

Understanding river flow regime dynamics is important for the management of ecosystems that
are sensitive to streamflow fluctuations. While climatic factors impact rainfall volumes during
wet and dry cycles, factors that control catchment runoff and baseflow are key to the
implementation of river protection strategies. In this study, groundwater components within
the J2000 model were distributed to improve baseflow and runoff proportioning for the
Verlorenvlei sub-catchment. The J2000 was distributed using groundwater model values for
the dominant baseflow tributary, while calibration was applied to the dominant streamflow
tributary. The model calibration was hindered by the maximum gauging station resolution,
which reduced the confidence in modelling high flow events, although an EMD filtering
protocol was applied to account for the resolution limitations and missing streamflow records.
The modelling approach would likely be transferable to other partially gauged semi-arid
catchments, provided that groundwater recharge is well constrained. The daily timestep nature
of the J2000 model allowed for an in-depth understanding of tributary flow regime dynamics,
showing that while streamflow variability is influenced by the runoff to baseflow proportion,
the host rock or sediment in which groundwater is held is also a factor that must be considered.
The modelling results showed that on average the streamflow influxes were not able to meet
the evaporation demand of the lake. High-flow events, although they occur infrequently, are
responsible for regeneration of the lake's water level and ecology, which illustrates the
importance of wet cycles in maintaining biodiversity levels in semi-arid environments. With
climate change likely to impact the length and occurrence of dry cycle conditions, wet cycles
are important for ecosystem regeneration.





## 8. Acknowledgements

The authors would like to thank the WRC and SASSCAL for project funding as well as the NRF and Iphakade for bursary support. Agricultural Research Council (ARC) and South African Weather Service (SAWS) for their access to climate and rainfall data.

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
