# Peer review of "Distributive rainfall/runoff modelling to understand runoff to baseflow"

_Hydrology and Earth System Sciences, 2018_

## Referee Comment (RC1) · Anonymous Referee #1 · 27 Sep 2018

Many of the references in the introduction are quite old, both those that refer to environmental flow requirement methods, as well as those referring to rainfall-runoff modelling approaches. I would have expected to see more references to the uncertainties inherent in hydrological modelling in a paper where there are limited gauging data to calibrate, assess and validate the model. The paper refers to Veloronvlei as both a lake and an estuary, which is it? How often is this water body linked to the sea and therefore how often is the water level influenced by sea water inputs? This is not mentioned in the paper at all apart from a passing reference to a sand bar. The paper also

makes no mention of whether the lake/estuary receives any direct inputs from ground-water and while this may not be the case, this issue should at least be addressed as part of the simulation of the water balance. The introduction refers to setting the ecological reserve, but it only becomes clear later that the paper is focused on the reserve for the lake/estuary and not for the rivers themselves. Equation 1 provides, what appears to be, the overall water balance equation for the model but makes no reference to a groundwater component. The way in which the model is described is incomplete and yet a lot of detail is given. A flow diagram would have helped and I had to go to the journal of hydrology paper to get a real sense of how the model actually works. Why is slope considered to be a major forcing parameter of recharge? Recharge is largely a vertical drainage process and would be influenced much more by the drainage characteristics of the material in the unsaturated zone than the surface slope of the topography. Section 4.1.2 indicates that the recharge estimates of the rainfall/runoff model were based on the estimates from MODFLOW – this is equivalent of calibrating one model against another and the validity of this approach needs to be further supported and the inherent uncertainties discussed. Line 372 suggest that the MODFLOW recharge values were 'validated' with J2000 recharge estimates. You cannot validate one model against another, all you can say is that the two models were in broad agreement. Page 16 (and elsewhere) refers to an apparently non-standard use of the Nash-Sutcliffe efficiency statistic and refers to Watson et al., 2018. However, in neither of these papers could I find a definition of these (E2 and E1) statistics. If they are not the standard statistic then they need to be defined. Nowhere in the paper could I find mention of how water use and its impacts on the gauge data (and the inflows into the lake from the ungauged catchment) have been taken into account in the model. At the same time, Google Earth clearly indicates that there is extensive water use in the catchment (centre pivots, farm dams etc.) that are likely to affect both surface water and groundwater dynamics. The indications are therefore that the model has been setup to represent natural conditions (i.e. ignoring water use), while it has been calibrated against an observed record that reflects water use (the same comment

applies to the earlier paper published in Journal of Hydrology). The presentation and discussion of the streamflow and baseflow results and other results (5.1 to 5.4) would have been clearer if presented in a table(s) supported by some explanatory text. I am surprised that the authors did not do some time series water balance modelling of the lake using the simulated inflows and reduced inflows to represent a 'reserve'. This would have avoided all the simplifications about an average evaporation loss. This would have been simple to do using a reservoir model. The reservoir model outputs using the simulated inputs could then have been converted to depths and compared with the observed depth data offering an additional method of assessing the model results. While some bathymetry data would be needed, I am sure some estimates could have been made, even if detailed bathymetry data are not available. Page 32 suggests that the 95th percentile is the ecological reserve percentile. This is simply not true. In South Africa (and most other countries) the reserve (or EWR) is expressed as a variable flow regime and never as a fixed FDC percentile. I am afraid that the whole discussion about the reserve indicates that the authors have little understanding of how reserves are estimated in a South African context.

Overall, the stated context of this paper is the determination of the ecological reserve, as supported by rainfall/runoff modelling. However, the environmental flow parts of the paper are far too simplified to justify publication. The paper therefore ends up being mostly focused on the hydrological model. However, there are not enough details provided in this paper to really assess the model or the results and heavy reliance is made on references to an earlier paper published in the Journal of Hydrology. I therefore do not see that this paper submitted to HESS adds anything new that is of scientific or practical relevance. Given that that are many other deficiencies in the manner in which this study has been conducted and presented, I have to recommend that this paper should not be published.

Other specific comments: Page 6: The paper refers to the catchment as a sub-catchment of the Olifants/Doorn quaternary catchment, but actually it is not in the

[Figure]

Olifants/Doorn catchment at all. Page 6: Reference is made to the lake supporting Karroid and Fynbos biomes, but these are terrestrial biomes that have no connection to any aquatic requirements. The paper also attributes the dual support of these biomes to the intermittent connection between salt and fresh water, which is clearly not correct and the salinity regime of the lake has nothing to do with the terrestrial biomes prevalent in the catchment. Page 6 mentions something about the salt and freshwater regimes of the lake, which will be critical to any environmental assessment, but no details are given and later in the paper this issue is totally ignored. Page 18: How do gauging station limitations result in good objective functions? This makes no sense to me. A casual glance at Fig 4 does not seem to support the conclusion that there were more gauge exceedance in the calibration period relative to the validation period. It would have been better to state how many were in each period. It is also not clear what the modellers did with these periods (set the observed data to missing values perhaps?). I could not find Table 1 in the submission. The title of Figure 6 does not seem to make grammatical sense? Page 28 says 'BFI values are generally below 1'. In fact BFI values are defined by baseflow/total flow and therefore are ALWAYS less than or equal to 1.

---

## Author Comment (AC1) · 2 Oct 2018

Dear Anonymous reviewer (#1)

We would like to thank the reviewer of our paper for their comments and feel that these will help in improving our paper. The aim of this paper is to provide the hydrological data or hydrological components required for ecological determination, and the paper has scientific contributions in that it deals with both distributive surface water and groundwater modelling as well as improving rainfall/runoff model baseflow separation

ability. The long time series data estimated for this sub-catchment is the first in this WMA and can be used for future water management as gauging structures and measurements are presently unavailable. While the reviewer has raised some issues that need to be addressed in the paper, there are some points that are not correct and some that are to do with the way in which the paper is written and not actually related to the results. We accept that parts of the manuscript can be written better, although the results themselves are sound and are vital for management of water resources in this water stressed region.

Regarding the references, we accept that these might be slightly out of date, although the catchment lacks site specific references which is part of the problem which compounds the need for research such as the one presented here. We have considered the uncertainty that limited data could have on hydrological models that are constructed for WMA such as Olifants/Doorn, although this has obviously not been brought through clearly enough in the introduction and we will update this accordingly. With regards to the Verlorenvlei system, this is classified as an estuarine-lake (e.g. Meadows et al., 1996 amongst many others) and we feel that this is clear in our manuscript. The sandbar limits the connection between the sea and freshwater. Water levels measured at G3T001 are significantly far away from the sandbar and therefore not impacted by coastal tides. Regarding the lake receiving direct inputs from groundwater, the daily evaporation rates in the sub-catchment are very high in which case groundwater contributions could be significant, although these volumes are not enough to counteract the daily evaporation potentials. While we do not consider this in the model setup, groundwater inflows into the lake could potentially be an important component of the water balance. While the focus of this contribution is quantifying stream and baseflow inputs into the lake using distributive modelling, in future we would like to be able to simulate the water levels in the lake as well as groundwater and surface water abstractions, although this will only be included in future studies.

The reviewer is correct that if you look at current Google Earth imagery of the catchment, there is clearly extensive agricultural development with many centre pivots which would impact streamflow. However, the model was calibrated between 1987 to 1993 (Fig. 7), when agricultural withdrawals from the catchment are far less intense (see Google Earth). Moreover, the model was calibrated for the Kruismans tributary (Fig 1), which has a far lower water footprint as the number of centre pivots are far less than the rest of the sub-catchment even today. Therefore, the calibration was conducted when river flow regimes where relatively unaltered and parameters estimated are valid for this sub-catchment. During the periods where there was no observed data (2007-2018), the data was set as missing values as presumed by the reviewer. An EMD protocol was applied to the water level data in the lake (2008-2018) to filter runoff data that resulted in a water level change at the sub-catchment outlet for further comparison, although further improvement can still be made to this approach.

Regarding Fig 4, streamflow exceeds the cut-off threshold of 3.675 m.3.s-1 for the station more frequently during the calibration, as this is during a wet cycle, with average rainfall of 413 mm/year as opposed to the dry cycle, which has an average of 330 mm/year (Fig. 4). We accept that a probability could be used to ascertain how often the cut-off threshold was exceeded, which would be more beneficial to the reader and can clarify this in the revised version. Reviewer 1 is correct that equation 1 refers to the overall water balance used, and it is not immediately clear how groundwater is part of the model presented, although equation 2 and 3 explain how recharge and interflow is determined, while equation 4 and 5 outline how slow and fast groundwater flow is calculated in the model. We accept that a flow diagram can be added to aid in this description and it will be included in the revised paper. Regarding the slope factor, this is a component of the J2000 model (Krause et al., 2005) and is a calibration factor used to determine the proportion of percolation to interflow (http://jams.uni-jena.de/ilmswiki/index.php/Hydrological_Model_J2000). It essentially represents the forces in the triangle of gravitation, normal force and frictional force. The latvertdist parameter is a representation of anisotropy, which modifies the forces triangle.

[Figure]

We agree that the MODFLOW estimates and the J2000 should not be calibrated to one another and that they should rather be in broad agreement with one another. Due to the J2000 lumping percolation rates (hydraulic conductivity), in this paper we have distributed the percolation rate for different geological formations by considering aquifer hydraulic and net recharge values that we used for a groundwater model calibrated for a sub-tributary. We believe this is just a wording issue and can be dealt with in the revised version. Whilst a reservoir model could be used to include the lake water level data thereby accounting for evaporation losses, further developments on the model structure still need to be made, which will be covered in future work. With regard to the catchment that the Verlorenvlei system sits in, it is quite clear that Verlorenvlei makes up the southern portion of the Olifants/Doorn Water Management Area (WMA). This is on a wide variety of published material, from Dept of Water Affairs maps to peer-reviewed journal articles and numerous consultancy reports. It is not clear to us why the reviewer thinks otherwise. Regarding the comments on page C4, these are minor and will be dealt with in the revised paper.

While this contribution uses a simplistic estimate for the ecological reserve, it provides the necessary data required for more comprehensive estimates and hence the need to make it open sourced. We accept that there might be some disparity as to how the paper conveys ecological reserve estimates which are quite out of date now, where Building Block Methodology (BBM) has become the standard for ecological reserve determination in South Africa. Therefore, this paper would require some minor restructuring to make this clearer. The distributive surface water modelling was covered in a previous paper published in Journal of Hydrology and therefore only a summary is included in this contribution. This is standard journal practice as most journals don't provide space for lengthy summaries of prior components of the same work. We will however, fine-tune the information presented to ensure that the most critical components are covered.
* * *
[Figure]

459, 2018.

---

## Referee Comment (RC2) · D.S. Stampoulis (Referee) · 30 Oct 2018

Reviewer: Dimitrios Stampoulis

Recommendation- Moderate revision

General Comments- In this paper the authors investigate river flow dynamics in order to enable a more efficient implementation of protection strategies and management for ecosystems that are sensitive to streamflow fluctuations. Specifically, streamflow variability and aquifer baseflow contributions in the Verlorenvlei lake system were assessed using the J2000 rainfall/runoff model, the groundwater components of which were distributed to improve baseflow and runoff proportioning for the aforementioned sub-catchment.

Overall, the work presented is significant, and the study is a rather considerable addition to the relative literature. The topic is within the scope of HESS, as it provides useful hydrological information that can potentially contribute towards achieving a more sustainable management of vulnerable ecosystems. The manuscript is generally very well-written. The methodological design is for the most part clear, however not entirely sound, and the authors' conclusions are well supported by their findings. Below, I outline a few general concerns, followed by a range of specific comments, which prevent me from recommending this manuscript for publication in its current form. I believe that the authors will be able to adequately address my comments and when that is done, this paper should be acceptable for publication.

Specific Comments-

1) The manuscript is not easy to read, due to the lack of a comprehensive structure that would help the reader easily understand the science and methodology. Please consider providing a more reader-friendly version of this paper, perhaps by changing the outline into a more compact one. 2) The authors needs to provide more information about the study area. Climatology-related information could be supported by a map or graph (time series). More detailed description about the regional hydrology is required. 3) Most of the references in the introduction are outdated. The authors need to make sure that they have conducted a thorough literature review. 4) The model is not sufficiently described. Please elaborate. 5) Are water abstractions taken into account by the model? It seems that this is not the case, and the authors need to clearly state this fact. 6) The results section is hard to read and follow; lack of supporting tables and graphs render reading a tedious task. The authors seem to have a lot of interesting results, which however, without a proper visualization have little meaning or use. Please consider using summarizing tables or time series or other graphs. 7)

Comparison between models is one thing, however one should not validate one model using the output of another. Please consider using an alternative data set or replace the word "validated" in Line 372 with "compared with". 8) The modeling approach is rather difficult to be transferred to other catchments as is, because of the different level of complexities in the geomorphological structure as well as the unique climatologies that characterize each specific region.

Technical Corrections-

1) Line 191 replace "was" with "were" 2) Lines 272-273 six or seven AWS's? 3) Line 361 Pbias 4) Line 497 In data-scarce

---

## Short Comment (SC1) · 9 Nov 2018

General comments:
The manuscript adapts a previously set-up surface water model by incorporating calibrated aquifer hydraulic conductivity values from a distributed groundwater model. The adapted model, representing the four main river tributaries upstream of a South African estuarine lake, is then used for calculating daily river- and base flows for each tributary during the period of 1987-2017. The model outputs are used for calculating tributary contributions, variability and flow exceedance percentiles – representing the

dynamics of this particular water system's flow regime. For instance, this information can be of use when determining the amount of water that should be reserved for ecological purposes (in the manuscript called 'the ecological reserve'). The results showed that the average streamflow influxes were not able to meet the approximated evaporative demand of the lake, highlighting the importance of high-flow events for regenerating the lake's water level and ecology.

I welcome model attempts to provide the data needed to maintain the ecological status of water systems. Indeed, this is particularly relevant for data scarce and ecologically sensitive areas. A comprehensive understanding of the flow dynamics, including both surface and subsurface flows, is needed to understand how to preserve ecological functions. To provide examples on how to combine results from two commonly used models (here MODFLOW and J2000) is also judged to be of value for the scientific community.

The intention of this manuscript is therefore relevant and important. In sum, I agree that the conclusions regarding the importance of high-flow events, illustrating the importance of wet cycles when maintaining biodiversity, is relevant. Nevertheless, I unfortunately have some major concerns with the manuscript. I would therefore like to encourage the authors to make som major revisions. This review will outline two main areas of improvement needs found with the manuscript, related to: 1) Methodology, and 2) Clarity.

Specific comments:
1) Methodology
1.1) The authors are encouraged to explain why a combination of these particular models are selected for this particular modelling challenge. What other options exists for combining distributed surface water modelling with distributed groundwater modelling?

1.2) The description of the study area states that agriculture is the dominant water user from the sub-catchment. But following this, the manuscript does not take agricultural expansion into account. Would not the increased irrigation in the area affect the streamflow data used for calibration, bringing non-stationary patterns? Is the water use taken into account in the model and how has water use developed during the simulation period? Also land-cover changes would be relevant to take into account, since the land cover data was only based on data from 2009. This is particularly important, to at least discuss, due to the fact that the manuscript presents agricultural expansion to be one of the major threats to the lake.

1.3) The estuarine nature of the lake is not taken into account or discussed. The evaporative demand is merely roughly approximated in the discussion. I wonder why this was not made with more care, since some of the main conclusions in the manuscript rely on this evaporative demand. The mix of salt and fresh water must mean that there is a dynamic flow exchange between the ocean and the lake. Please consider to explain/discuss why is this not taken into account in the modelling.

1.4) The authors are suggested to explain why the two areas for surface and sub-surface calibration was selected. The authors are also encouraged to discuss the implications of the calibration and validation limitations, especially in relation to the major calibration data gap and the fact that the measuring gauge had an upper measurement limitation.

2) Clarity

2.1) One of the main issues with this manuscript is its unclear aim. This is suggested to be written in a more clear and concise way. Following from this, the distinction between general information (e.g. model equations in the J2000 model), previously done work and the novelty of this particular manuscript becomes fuzzy. The authors are strongly recommended to make this clearer. For instance, when describing the water balance calculations in chapter 3.3, it is also necessary to clarify what information is general for the software used and what is specifically chosen for this study.

2.2) A majority of the chapters would benefit from being written more concise and to-the-point. General model information could be left out with reference to the model documentation, the model settings and the results would benefit from being presented in tables.

2.3) Chapter 4.2.2 is describing the surface water calibration. But the section describing the groundwater calibration is missing (or possibly it is just the headline that is missing). This is a gap that is suggested to be highly relevant for this manuscript.

Technical corrections:

-It is difficult to distinguish the colours in the hydrogeological map in Figure 2.

-The text description of Figure 4 has an error; the period of validation should be for 1994-2006.

-I would encourage the authors to more carefully describe why this particular lake system was selected for the case study. The manuscript states that the estuarine system is "under threat from climate change and agricultural expansion". The term "under threat" is vague, and the statement is not referenced.

-The authors are referencing to their own unpublished work. This reference is furthermore not included in the reference list. This is problematic with regard to transparency, since no access to this source is given. The authors are encouraged to consider other ways of providing this information, for instance through supplementary materials (if possible).

-There is a general issue of missing references, for instance the geological data in chapter 2 and the parameter values in chapter 3.3.2. The reference list needs to be revised, at least one reference is missing (Sigidi, 2018).

-The appropriate number of significant figures should be revised. It is not reasonable to give exceedance percentiles with six significant figures (Table 3), due to uncertainties and limitations in input data and models.

-The headline for chapter 4 is missing.

---

## Author Comment (AC2) · 7 Dec 2018

On behalf of myself and all the co-authors I would like to thank all the reviewers for commenting on our manuscript, your comments are much appreciated and will aid in improving our paper. The reviewers have highlighted a few concerns and additions that we can address in a revised version and this response aids in describing the changes that will be made. Both reviewers highlighted the need to improve the structure of the paper to make it clearer, as well as making it more concise and highlighting its novelty. We agree that the references are somewhat out of date, which was also a comment
from the first reviewer, which we will revise. While Figure 1 shows the MAP of the study catchment we agree that there might be some additional representations that could help the readership understand the catchment conditions better. While a comprehensive breakdown of the environmental setting exists in another, cited paper in Journal of Hydrology (Watson et al., 2018) we agree that more background information is required so that this paper can be a standalone contribution and not require the reader to review to previous papers. As such we will address this by improving this section, including more information which is relevant to understand the catchment conditions. Similar to the study catchment comment, the description of the model in this contribution is somewhat reliant on the previous Journal of Hydrology paper, which was also a concern for the first reviewer and which will be addressed. In terms of abstraction, while this inclusion is possible in the case of MODFLOW, it is not possible with the coupled J2000 setup at present and is a future interest for us. As such we will make this clearer in the model description, outlining this limitation. There is enough information to include the sub-catchment irrigation and this will be included with a description of the method as well as the impact on the water balance. We agree that a more up to date landcover dataset would be more representative if an active gauging structure existed, although as the gauging data is between 1987-2008, the 2007 landcover dataset is better for the model calibration. A 2013/2014 National Land Cover dataset exists for South Africa, which we will incorporate in future models once the initial model approach has been completed. The evaporation from the lake was initially not included as the sea water influx from the coast is unquantifiable at present, making it difficult to separate out the fresh water evaporation in the model. Although, with remote sensing data and measured lake water levels, lake evaporation processes can be represented in J2000, improving the lake ET estimates and thereby quantifying how frequently the influx from the feeding tributaries meets the freshwater evaporation demand. As such the revised version will incorporate the lake ET package for estimating the evaporation off the lake, as well as simulating riparian reed ET which could also have an impact on lake water levels. The J2000 was calibrated between 1987-1994 for the Kruismans tributary, as

the gauging station existed on this tributary. The groundwater validation was applied to the Krom Antonies as it was initially believed the most dominant source of baseflow. We accept that this has not come out clear enough in the paper and adjustments will be made. The J2000-MODFLOW was used as opposed to other models as the J2000 has the capability to simulate and represent certain processes required to understand the tributary river flow regime as well as it being a fully distributed model. In addition to this, as the Verlorenvlei is somewhat partially gauged, we used parameter values that were determined for an adjacent sub-catchment (Sandspruit), which helped to narrow the J2000 calibration values. We agree that this information is required for the readership and will be revised. One reviewer has some concerns about how the results have been presented, in particular to the graphs presented, we accept that as there was quite a large amount of data to present, it might be a bit confusing to the reader and we will revise this. There remain some additions that can be applied to the discussion and conclusion which the reviewer has highlighted, where the application to other regions is important for the readership. We accept that there are some minor technical corrections required to the paper, which we will address in the revised version. In conclusion, the revisions outlined above will not substantially change the findings of the paper, therefore this contribution should be suitable for publication in HESS.
* * *

---

## Author Response (AR1)

**Distributive rainfall/runoff modelling to understand runoff to baseflow proportioning and its impact on the determination of reserve requirements of the Verlorenvlei estuarine lake, west coast, South Africa**

Andrew Watson1, Jodie Miller1, Manfred Fink2, Sven Kralisch2, Melanie Fleischer2, and Willem de Clercq3

Reviewer 1: Overall, the stated context of this paper is the determination of the ecological reserve, as supported by rainfall/runoff modelling. However, the environmental flow parts of the paper are far too simplified to justify publication." And "I am surprised that the authors did not do some time series water balance modelling of the lake using the simulated inflows and reduced inflows to represent a 'reserve'."

**Please note change of title to due comments from reviewer 1**

**Reviewer 1: Anonymous**

1) Many of the references in the introduction are quite old, both those that refer to environmental flow requirement methods, as well as those referring to rainfall-runoff modelling approaches. I would have expected to see more references to the uncertainties inherent in hydrological modelling in a paper where there are limited gauging data to calibrate, assess and validate the model. **Response-** Accepted changes made: **References have updated with more relevant literature, although no changes make in regard to referring runoff-model limitation and inclusion of why the models were chosen has been made.**

2) The paper refers to Veloronvlei as both a lake and an estuary, which is it? How often is this water body linked to the sea and therefore how often is the water level influenced by sea water inputs? This is not mentioned in the paper at all apart from a passing reference to a sand bar. **Response-** Accepted changes made: **Its an estuarine lake,** (Sinclair et al., 1986), "A sandbar created around a sandstone outcrop (Table Mountain Group) allows for an intermittent connection between salt and fresh water. During storms or extremely high tides, water scours the sand bar allowing for a tidal exchange, with a constant inflow of salt water continuing until the inflow velocity decreases enough for a new sand bar to form (Sinclair et al., 1986)" Line 129-133. **I am afraid there is little information about the coastal exchange, and is something we want to look at into the future as it has a bearing on the lake water level.**

3) The paper also makes no mention of whether the lake/estuary receives any direct inputs from groundwater and while this may not be the case, this issue should at least be addressed as part of the simulation of the water balance.

Response-Rejected paper unchanged: Regarding the lake receiving direct inputs from groundwater, the daily evaporation rates in the sub-catchment are very high in which case groundwater contributions could be significant, although these volumes are not enough to counteract the daily evaporation potentials.

4) The introduction refers to setting the ecological reserve, but it only becomes clear later that the paper is focused on the reserve for the lake/estuary and not for the rivers themselves. Response-Accepted changes made:Revised paper talks about the reserve as opposed to ecological reserve of each river 5) Equation 1 provides, what appears to be, the overall water balance equation for the model but makes no reference to a groundwater component.

**Response-** Rejected paper unchanged:

Reviewer 1 is correct that equation 1 refers to the overall water balance used, and it is not immediately clear how groundwater is part of the model presented, although equation 2 and 3 explain how recharge and interflow is determined, while equation 4 and 5 outline how slow and fast groundwater flow is calculated in the model.

6) The way in which the model is described is incomplete and yet a lot of detail is given. A flow diagram would have helped and I had to go to the journal of hydrology paper to get a real sense of how the model actually works.

**Response- Accepted changes made: A flow diagram has been included which shows the processed followed by the modelling**

7) Why is slope considered to be a major forcing parameter of recharge? Recharge is largely a vertical drainage process and would be influenced much more by the drainage characteristics of the material in the unsaturated zone than the surface slope of the topography.

Response-Rejected paper unchanged: Regarding the slope factor, this is a component of the J2000 model (Krause et al., 2005) and is a calibration factor used to determine the proportion of percolation to interflow (http://jams.unijena. de/ilmswiki/index.php/Hydrological\_Model\_J2000). It essentially represents the forces in the triangle of gravitation, normal force and frictional force. The latvertdist parameter is a representation of anisotropy, which modifies the forces triangle.

Section 4.1.2 indicates that the recharge estimates of the rainfall/runoff model were based on the estimates from MODFLOW – this is equivalent of calibrating one model against another and the validity of this approach needs to be further supported and the inherent uncertainties discussed. Line 372 suggest that the MODFLOW recharge values were 'validated' with J2000 recharge estimates. You cannot validate one model against another, all you can say is that the two models were in broad agreement.

**Response- Accepted changes made: Adjustments have been made throughout the paper to align with the comments above.**

Reviewer 1: Page 16 (and elsewhere) refers to an apparently non-standard use of the Nash-Sutcliffe efficiency statistic and refers to Watson et al., 2018. However, in neither of these papers could I find a definition of these (E2 and E1) statistics. If they are not the standard statistic then they need to be defined.

Response- Rejected. The Nash-Sutcliffe presented is in standard form of efficiency and will not be discussed in this contribution. Refer to Nepal, S., 2012 for more on this.

Nowhere in the paper could I find mention of how water use and its impacts on the gauge data (and the inflows into the lake from the ungauged catchment) have been taken into account in the model. **Response-**Noted, changes made. "During winter, the majority of the irrigation water needed for crop growth is supplied by the sub-catchment tributaries or lake itself, although the impact of irrigation is still regarded as minimal (Meinhardt et al., 2018) and requires future investigation" **While this is a valid comment, the incorporation of irrigation and groundwater abstraction is a future interest and when the data is available this will be incorporated in a future contribution.**

At the same time, Google Earth clearly indicates that there is extensive water use in the catchment (centre pivots, farm dams etc.) that are likely to affect both surface water and groundwater dynamics. The indications are therefore that the model has been setup to represent natural conditions (i.e. ignoring water use), while it has been calibrated against an observed record that

reflects water use (the same comment applies to the earlier paper published in Journal of Hydrology).

Response-Rejected, no changes made. The reviewer is correct that if you look at current Google Earth imagery of the catchment, there is clearly extensive agricultural development with many centre pivots which would impact streamflow. However, the model was calibrated between 1987 to 1993 (Fig. 7), when agricultural withdrawals from the catchment are far less intense (see Google Earth). Moreover, the model was calibrated for the Kruismans tributary (Fig 1), which has a far lower water footprint as the number of centre pivots are far less than the rest of the subcatchment even today. Therefore, the calibration was conducted when river flow regimes where relatively unaltered and parameters estimated are valid for this sub-catchment. During the periods where there was no observed data (2007-2018), the data was set as missing values as presumed by the reviewer.

The presentation and discussion of the streamflow and baseflow results and other results (5.1 to 5.4) would have been clearer if presented in a table(s) supported by some explanatory text. I am surprised that the authors did not do some time series water balance modelling of the lake using the simulated inflows and reduced inflows to represent a 'reserve'. This would have avoided all the simplifications about an average evaporation loss. This would have been simple to do using a reservoir model. The reservoir model outputs using the simulated inputs could then have been converted to depths and compared with the observed depth data offering an additional method of assessing the model results. While some bathymetry data would be needed, I am sure some estimates could have been made, even if detailed bathymetry data are not available. **Response-** Accepted changes made: **New figures have been included in the results section with the model incorporating lake ET.**

Page 32 suggests that the 95th percentile is the ecological reserve percentile. This is simply not true. In South Africa (and most other countries) the reserve (or EWR) is expressed as a variable flow regime and never as a fixed FDC percentile. I am afraid that the whole discussion about the reserve indicates that the authors have little understanding of how reserves are estimated in a South African context.

Response- Accepted changes made: Changes have been made to the paper so that the new version does not conflict with how ecological reserves are determined in South Africa.

Overall, the stated context of this paper is the determination of the ecological reserve, as supported by rainfall/runoff modelling. However, the environmental flow parts of the paper are far too simplified to justify publication. The paper therefore ends up being mostly focused on the hydrological model. However, there are not enough details provided in this paper to really assess the model or the results and heavy reliance is made on references to an earlier paper published in the Journal of Hydrology

Response- Accepted changes made: As per above comment.

Other specific comments: Page 6: The paper refers to the catchment as subcatchment of the Olifants/Doorn quaternary catchment, but actually it is not in the Olifants/Doorn catchment at all. Response-Noted, Changes made: With regard to the catchment that the Verlorenvlei system sits in, it is quite clear that Verlorenvlei makes up the southern portion of the Olifants/Doorn Water Management Area (WMA). This is on a wide variety of published material, from Dept of Water Affairs maps to peer reviewed journal articles and numerous consultancy reports. It is not clear to us why the reviewer thinks otherwise.

Reviewer 1: Page 6: Reference is made to the lake supporting Karroid and Fynbos biomes, but these are terrestrial biomes that have no connection to any aquatic requirements. The paper also attributes the dual support of these biomes to the intermittent connection between salt and fresh water, which is clearly not correct and the salinity regime of the lake has nothing to do with the terrestrial biomes prevalent in the catchment. Page 6 mentions something about the salt and freshwater regimes of the lake, which will be critical to any environmental assessment, but no details are given and later in the paper this issue is totally ignored

**Response-** Accepted changes made:" The estuarine lake hosts both Karroid and Fynbos biomes, with a variety of vegetation types (e.g Arid Estuarine Saltmarsh, Cape Inland Salt pans) being sensitive to reduced inflow of freshwater (Helme, 2007)"

Page 18: How do gauging station limitations result in good objective functions? This makes no sense to me. A casual glance at Fig 4 does not seem to support the conclusion that there were more gauge exceedance in the calibration period relative to the validation period. It would have been better to state how many were in each period. It is also not clear what the modellers did with these periods (set the observed data to missing values perhaps?).

**Response- Noted**

Regarding Fig 4, streamflow exceeds the cut-off threshold of 3.675 m.3.s-1 (DT limit) for the station more frequently during the calibration, as this is during a wet cycle, with average rainfall of 413 mm/year as opposed to the dry cycle, which has an average of 330 mm/year (Fig. 4). We accept that a probability could be used to ascertain how often the cut-off threshold was exceeded but this is not the objective of this contribution and would make the already lengthened revised version far longer than intended.

I could not find Table 1 in the submission. Response- Reject, no changes made: Page 14

The title of Figure 6 does not seem to make

Page 28 says 'BFI values are generally below 1'. In fact BFI values are defined by baseflow/total flow and therefore are ALWAYS less than or equal to 1.

Response- Reject, no changes made: As this paper is focused on understanding river flow regime dynamics, this is particular important for the readership, and while it might seem like it will always be less than 1, CV values are spoken in the same line and require a range for understanding. In that same vain, if its obvious then why have other articles stated this?

**Reviewer 2: D.S. Stampoulis**

1) The manuscript is not easy to read, due to the lack of a comprehensive structure that would help the reader easily understand the science and methodology. Please consider providing a more reader-friendly version of this paper, perhaps by changing the outline into a more compact one **Response-** Accepted changes made: **The manuscript structure ahs been revised, in particular the methodology has been improved to be easier to follow.**

2) The authors needs to provide more information about the study area. Climatology-related information could be supported by a map or graph (time series). More detailed description about the regional hydrology is required.

Response- Accepted changes made: New figure included which shows how the rainfall has varied for the last 52 years. This leads onto how rainfall varies spatially across the catchment, with a significant different between the valley and mountains and the different geological formations.

3) Most of the references in the introduction are outdated. The authors need to make sure that they have conducted a thorough literature review.

Response- Accepted changes made: As per reviewer 1 changes have been made to the references

4) The model isnot sufficiently described. Please elaborate.

Response- Accepted changes made: New flow chart has been included to describe how the model works.

5) Are water abstractions taken into account by the model? It seems that this is not the case, and the authors need to clearly state this fact.

Response- Accepted changes made: Please see method section, where a new flow diagram and introduction sentence has been implemented to address this. Final discussion section is also regarding the irrigation and what impact it could have.

6) The results section is hard to read and follow; lack of supporting tables and graphs render reading a tedious task. The authors seem to have a lot of interesting results, which however, without a proper visualization have little meaning or use. Please consider using summarizing tables or time series or other graphs.

Response- Accepted changes made: The result figures have been revised, with two new figures which a pie chart of the flow contributions and flow component proportions.

Comparison between models is one thing, however one should not validate one model using the output of another. Please consider using an alternative data set or replace the word "validated" in Line 372 with "compared with".

Response- Accepted changes made: As per reviewer 1, changes made throughout to align with this.

8) The modeling approach is rather difficult to be transferred to other catchments as is, because of the different level of complexities in the geomorphological structure as well as the unique climatologies that characterize each specific region.

Response- Accepted changes made: Further developments have been made to the discussion which look at how the impacts of dry and wet cycles could impact sensitive ecosystems such as the Verlorenvlei.

Technical Corrections-1) Line 191 replace "was" with "were" 2) Lines 272-273 six or seven AWS's? 3) Line 361 Pbias 4) Line 497 In data-scarce Response- Accepted changes made: **Changes made.**

**Short comment: S. Andersson**

1.1) The authors are encouraged to explain why a combination of these particular models are selected for this particular modelling challenge. What other options exists for combining distributed surface water modelling with distributed groundwater modelling?

**Response-** Accepted changes made: "To better understand river flow variability, a rainfall/runoff model was distributed to incorporate aquifer hydraulic conductivity within model HRUs using calibrated values from a MODFLOW groundwater model (Watson, 2018). The rainfall/runoff model

used was J2000 as this model had previously been set up in the region and model variables were well established (e.g Bugan, 2014; Schulz et al., 2013)".

1.2) The description of the study area states that agriculture is the dominant water user from the sub-catchment. But following this, the manuscript does not take agricultural expansion into account. Would not the increased irrigation in the area affect the streamflow data used for calibration, bringing non-stationary patterns? Is the water use taken into account in the model and how has water use developed during the simulation period? Also land-cover changes would be relevant to take into account, since the land cover data was only based on data from 2009. This is particularly important, to at least discuss, due to the fact that the manuscript presents agricultural expansion to be one of the major threats to the lake.

**Response-** Accepted changes made: "Agriculture is the dominant water user in the sub-catchment with an estimated usage of 20 % of the total recharge (DWAF, 2003; Watson, 2018), with the main food crop being potatoes. The MG shales and quaternary sediments, which host the secondary and primary aquifer respectfully, are frequently used to supplement irrigation during the summer months of the year. During winter, the majority of the irrigation water needed for crop growth is supplied by the sub-catchment tributaries or the lake itself. The impact of irrigation on the lake is still regarded as minimal (Meinhardt et al., 2018) but requires future investigation." We agree that a more up to date landcover dataset would be more representative if an active gauging structure existed, although as the gauging data is between 1987-2008, the 2007 landcover dataset is better for the model calibration. A 2013/2014 National Land Cover dataset exists for South Africa, which we will incorporate in future models once the initial model approach has been completed.

1.3) The estuarine nature of the lake is not taken into account or discussed. The evaporative demand is merely roughly approximated in the discussion. I wonder why this was not made with more care, since some of the main conclusions in the manuscript rely on this evaporative demand. The mix of salt and fresh water must mean that there is a dynamic flow exchange between the ocean and the lake. Please consider to explain/discuss why is this not taken into account in the modelling. **Response-** Accepted changes made: **Lake ET has been incorporated in the new model. There still remains very little information regarding the sea water exchange and this remains something we would like to address in future papers.**

1.4) The authors are suggested to explain why the two areas for surface and subsurface calibration was selected. The authors are also encouraged to discuss the implications of the calibration and validation limitations, especially in relation to the major calibration data gap and the fact that the measuring gauge had an upper measurement limitation.

Response-Noted. The surface water calibration was applied to the Kruismans, which is the only tributary with streamflow measurements (This has been stated in the paper). The Krom Antonies was used for the groundwater component as it was believed the most significant in terms of baseflow (Stated in the paper).

**2) Clarity**

2.1) One of the main issues with this manuscript is its unclear aim. This is suggested to be written in a more clear and concise way. Following from this, the distinction between general information (e.g. model equations in the J2000 model), previously done work and the novelty of this particular manuscript becomes fuzzy. The authors are strongly recommended to make this clearer. For

instance, when describing the water balance calculations in chapter 3.3, it is also necessary to clarify what information is general for the software used and what is specifically chosen for this study. Response- Accepted changes made: New flow diagram included which should clarify this issue and restructuring of the method and results sections

2.2) A majority of the chapters would benefit from being written more concise and to-the-point. General model information could be left out with reference to the model documentation, the model settings and the results would benefit from being presented in tables.

Response- Accepted changes made: Results have been revised as well as method.

2.3) Chapter 4.2.2 is describing the surface water calibration. But the section describing the groundwater calibration is missing (or possibly it is just the headline that is missing). This is a gap that is suggested to be highly relevant for this manuscript.

**Response-** Accepted changes made: **Heading missing, changes made.**

Technical corrections: -It is difficult to distinguish the colours in the hydrogeological map in Figure 2.

Response- Accepted changes made

-The text description of Figure 4 has an error; the period of validation should be for 1994-2006.

Response- Accepted changes made

-I would encourage the authors to more carefully describe why this particular lake system was selected for the case study. The manuscript states that the estuarine system is "under threat from climate change and agricultural expansion". The term "under threat" is vague, and the statement is not referenced.

**Response-** Accepted changes made: "The Verlorenvlei lake, which is approximately 15 km2 in size draining a watershed of 1832 km2, forms the southern sub-catchment of the Olifants/Doorn water management area (WMA). The estuarine lake hosts both Karroid and Fynbos biomes, with a variety of vegetation types (e.g Arid Estuarine Saltmarsh, Cape Inland Salt pans) being sensitive to reduced inflow of freshwater (Helme, 2007). A sandbar created around a sandstone outcrop (Table Mountain Group) allows for an intermittent connection between salt and fresh water. During storms or extremely high tides, water scours the sand bar allowing for a tidal exchange, with a constant inflow of salt water continuing until the inflow velocity decreases enough for a new sand bar to form (Sinclair et al., 1986). "

-The authors are referencing to their own unpublished work. This reference is furthermore not included in the reference list. This is problematic with regard to transparency, since no access to this source is given. The authors are encouraged to consider other ways of providing this information, for instance through supplementary materials (if possible).

Response- Accepted changes made

-There is a general issue of missing references, for instance the geological data in chapter 2 and the parameter values in chapter 3.3.2. The reference list needs to be revised, at least one reference is missing (Sigidi, 2018).

**Response- Accepted changes made**

-The appropriate number of significant figures should be revised. It is not reasonable to give exceedance percentiles with six significant figures (Table 3), due to uncertainties and limitations in input data and models.

Response-Accepted changes made. Although for flow exceedances it is not possible to use 2 significant figures otherwise streamflow is below 0.00 when in m3.s-1

-The headline for chapter 4 is missing. **Response**-Accepted changes made

Please see attached below for changes made to the revised paper. As there were multiple inputs from various authors track changes became very messy and hard to follow, especially with the addition of new diagrams and therefore were not possible in this response. I have highlighted major sections that were revised to show how this revised version differs from the original, although small changes that were incorporated have not been highlighted and require you to refer to the new manuscript.

[revised manuscript text omitted]

---

## Referee Report (RR1)

**Review of hess-2018-459 "Distributive rainfall/runoff modelling to understand runoff to baseflow proportioning and its impact on the determination of reserve requirements of the Verlorenvlei estuarine lake, west coast, South Africa" by Andrew Watson; Jodie Miller; Manfred Fink; Sven Kralisch; Melanie Fleischer; Willem de Clercq**

**Reviewer: Dimitrios Stampoulis**

**Recommendation-** Accepted subject to technical corrections

**General Comments**- I am still not satisfied with the literature review the authors have conducted, as on the one hand one citation (one of the authors' recently published article) is repeatedly used in many cases, while on the other hand I believe that the authors can do a better job further enhancing their literature review. Nevertheless, all of my previous concerns have been adequately addressed by the authors, and the paper should now be acceptable for publication, following minor revision, focusing on the aforementioned comment and the minor corrections indicated below:

**Technical Corrections-**

1) Line 30 remove "the" after "Of these,"
2) Line 33 same after "Instead," and after "baseflow."
3) Line 43 The length of dry cycles is likely to
4) Lines 54-56 Awkward sentence; please rephrase
5) Lines 88, 89 "set up" and not "setup"
6) Line 157 remove "the" before "Krom Antonies" – the authors need to make sure that this will be done for all similar cases throughout the text
7) Line 167 I do not have expertise in the relative field, but perhaps MG could be defined here
8) Line 557 remove "on"
9) Line 674 modeling

---

## Author Response (AR2)

**Dimitrios Stampoulis**

I am still not satisfied with the literature review the authors have conducted, as on the one hand one citation (one of the authors' recently published article) is repeatedly used in many cases, while on the other hand I believe that the authors can do a better job further enhancing their literature review. Nevertheless, all of my previous concerns have been adequately addressed by the authors, and the paper should now be acceptable for publication, following minor revision, focusing on the aforementioned comment and the minor corrections indicated below:

**Accept, we have made an adjustment and only cited previous work when necessary and**

**use other references in other cases. Additional references have been included to support**

**the literature review.**

Technical Corrections-

1) Line 30 remove "the" after "Of these,"

**Accept**

2) Line 33 same after "Instead," and after "baseflow."

**Accept**

3) Line 43 The length of dry cycles is likely to

**Accept**

4) Lines 54-56 Awkward sentence; please rephrase

**Accept, as per reviewer 2 comments**

5) Lines 88, 89 "set up" and not "setup"

**Accept**

6) Line 157 remove "the" before "Krom Antonies" – the authors need to make sure that this will be done for all similar cases throughout the text

**Accept, changes made throughout the paper**

7) Line 167 I do not have expertise in the relative field, but perhaps MG could be defined here

**Accept, MG defined line 146**

8) Line 557 remove "on"

**Accept**

9) Line 674 modeling

**Reject: In South Africa its modelling.**

**Anonymous**

The authors have modified the J2000 model to understand the contributions of the different flow components to the Verlorenvlei lake, South Africa from the various contributing tributaries. Overall, the study presents a significant body of work which adds to the available literature in this area. Furthermore, the work is of value to South African hydrology, where studies such of this that determine groundwater contributions together with surface water contributions are rare.

The manuscript in the revised form, is well written, the methodology clear and the conclusions generally supported by the findings. The authors have to a large extent addressed the comments of the previous reviewers and significantly improved the paper. The figures and tables that have been included in the revised version have improved the clarity of the paper.

It is my opinion that the paper still only touches on the ecological reserve aspects. The paper is a sound contribution to modelling and the determination of the components of flow, by attempting to relate to the ecological reserve the authors have detracted from the work as they have not adequately addressed it.

**Noted, in the online peer review, concerns were made by an anonymous reviewer**

**regarding the ecological reserve, stating that the methodology followed was not inline**

**with South African ecological reserve assessments. Subsequently in the revisions we**

**withdrew some of the initial discussion points and conclusions, so that the revised version**

**is both inline with South African ecological reserve assessments as well as relating the**

**work presented to the need for better estimates to ensure sustainable water usage.**

The authors have used MODFLOW to validate the recharge estimates for the J2000 model (Section 3.7.3; figure 7). Models should not be validated against each other, at most their outputs can be compared. I would encourage the authors to use another dataset for the calibration and validation, or to better document the pitfalls and uncertainties introduced with this approach and why it was the only one possible.

**Accept, this was also a concern for reviewer 1 in the original paper, but we have included**

**additional references to validate the procedure.**

"This was done by aligning the MODFLOW recharge estimates and previous studies (Conrad et al., 2004; Miller et al., 2017; Vegter, 1999; Weaver et al., 1999; Wu, 2005) with those of the

J2000, through adjustment of aquifer hydraulic conductivity from the MODFLOW

groundwater model of Krom Antonies (Watson, 2018) (Fig. 5)".

**"Recharge estimates from previous studies of the primary aquifer indicate recharge rates**

**of 0.2-3.4 % (Conrad et al., 2004), and 8% (Vegter, 1999), while for the TMG aquifer 13**

**% (Wu, 2005), 27% (Miller et al., 2017) and 17.4 % (Weaver and Talma, 2005) of MAP."**

The flows (Figure 9) from all tributaries are significantly higher during the last wet period (2007 – 2017). Please note that the average rainfall values used in the text and those in added in Figure 9 are not the same and should be corrected. The findings show higher flows in the latter period that are far greater than what would be anticipated from the higher rainfall. A 30

mm change in rainfall resulted in a more than doubling of the average baseflow response. It is during the latter period that the authors state that irrigation in the catchment has been expanding and that there is this growing threat of agriculture expansion to the water resources, but this has not been accounted for in the model. Have the authors considered the cause of the marked higher streamflow response – is it a change in the nature of the rainfall distribution in the latter period, a change in the timing of the rainfall?

**Accept, this is absolutely true. We looked into the model results for the first wet cycle and**

**the second, and saw a marked increase in soil moisture, with a minor decrease in potential**

**ET. Looking into the standard deviation between yearly rainfall for the first wet cycle,**

**the dry cycle and the second wet cycle there is more than a doubling in the yearly rainfall**

**variability, which is the result of this high flow variability in the second wet cycle. We**

**have incorporated the STD for the three cycles in the text and a small sentence in the**

**discussion about this, as the paper length has got quite long.**

"The estimated flow exceedance probabilities indicated that during the 2008-2017 wet cycle average lake inflows exceeded the average evaporation demand, **although yearly rainfall is twice as variable in comparison to the first wet cycle between 1987-1996". Line 35-38**

"This is particularly evident in the measured water level data from station G3T001, where measured water levels have a large daily standard deviation (0.62) (Watson et al., 2018). **The daily inflows of water into the Verlorenvlei has also been subject to significant rainfall variability, with yearly rainfall between the first wet cycle (1987-1996) being twice as variable in comparison to the second wet cycle (2007-2017). The change in rainfall variability has had a significant impact on soil moisture conditions, resulting in not only larger peak discharges but also lengthened low flow conditions.** With climate change likely to impact the length and severity of dry cycles, it is likely that the lake will dry up more frequently into the future, which could have severe implications on the biodiversity that relies on the lake's habitat for survival. Of importance to the lake's survival is the protection of river inflows during wet cycles, where the lake requires these inflows for regeneration". **Line 649-659**

The modelling results showed that on average the streamflow influxes were not able to meet the evaporation demand of the lake, **with yearly rainfall becoming more variable. Line 689-691**

Specific comments:

Introduction, Pg 3, Line 54/55 – insert "were" between "problems thought"

**Accept, as per reviewer 1 request**

Caption Fig 1. Isohyets spelt incorrectly

**Accept**

Section 2, Pg 8, Line 153 – The sentence "Where rainfall was less than 50 % of the MAP

(1965-1969 and 2015-2017), concerns over the amount of streamflow required to support the lake have been raised." Consider rephrasing this sentence. I would presume that the concerns mentioned are now for the recent past not the 1960's. You have stated that agric expansion has been a more recent phenomena in the catchment and is probably a driver of the concerns.

**Accept,** "Recently, where rainfall was less than 50 % of the MAP (2015-2017), concerns over the amount of streamflow required to support the lake have been raised"

Table 2, what are the units of AVE?

**Noted, absolute sum of differences between measured and simulated**, "absolute volume error (AVE)"

[revised manuscript text omitted]